# Act to See, See to Act: Diffusion-Driven Perception-Action Interplay for Adaptive Policies

**Jing Wang**[1]    **Weiting Peng**[2]    **Jing Tang**[2]    **Zeyu Gong**[2]    **Xihua Wang**[1]
**Bo Tao**[2]    **Li Cheng**[1]

[1]University of Alberta        [2]Huazhong University of Science and Technology
{jing39, xihua, lcheng5}@ualberta.ca
{u202210565, j_tang, gongzeyu, taobo}@hust.edu.cn

## Abstract

Existing imitation learning methods decouple perception and action, which overlooks the causal reciprocity between sensory representation and action execution that humans naturally leverage for adaptive behaviors. To bridge this gap, we introduce Action-Guided Diffusion Policy (DP-AG), a unified representation learning that explicitly models a dynamic interplay between perception and action through probabilistic latent dynamics. DP-AG encodes latent observations into a Gaussian posterior via variational inference and evolves them using an action-guided SDE, where the Vector–Jacobian Product (VJP) of the diffusion policy's noise predictions serves as a structured stochastic force driving latent updates. To promote bidirectional learning between perception and action, we introduce a cycle-consistent contrastive loss that organizes the gradient flow of the noise predictor into a coherent perception–action loop, enforcing mutually consistent transitions in both latent updates and action refinements. Theoretically, we derive a variational lower bound for the action-guided SDE, and prove that the contrastive objective enhances continuity in both latent and action trajectories. Empirically, DP-AG significantly outperforms state-of-the-art methods across simulation benchmarks and real-world UR5 manipulation tasks. As a result, our DP-AG offers a promising step toward bridging biological adaptability and artificial policy learning. Code is available on our project website: `https://jingwang18.github.io/dp-ag.github.io/`.

## 1    Introduction

Imitation learning (IL) enables agents to replicate expert behavior from demonstrations. Direct mapping methods, such as [Codevilla et al., 2018, Mandlekar et al., 2022, Florence et al., 2022], learn a direct mapping between observations and actions. In contrast, generative models such as Diffusion Policy (DP) [Chi et al., 2023] and flow-matching methods [Hu et al., 2024, Zhang et al., 2025] model action distributions to improve continuity across time steps. Vision-Language-Action (VLA) [Kim et al., 2024, Black et al., 2024] improves perception by leveraging Vision-Language Models (VLMs) to interpret environmental cues and high-level instructions. Despite progress, existing methods treat observation features as static during each action sequence generation (extracting them from a single time-point observation and holding them fixed while generating a short sequence of actions), thus overlooking the opportunity for the intermediate action feedback to refine perception.

Robust decision-making relies on a continuous interplay between perception and action [O'regan and Noë, 2001]. Humans naturally embody this principle by dynamically refining their environmental understanding through feedback from their own actions [Brooks, 1991]. Motivated by this, we propose *Action-Guided Diffusion Policy* (DP-AG), a representation learning framework for IL that explicitly models the perception–action interplay through probabilistic latent dynamics. We build upon DP

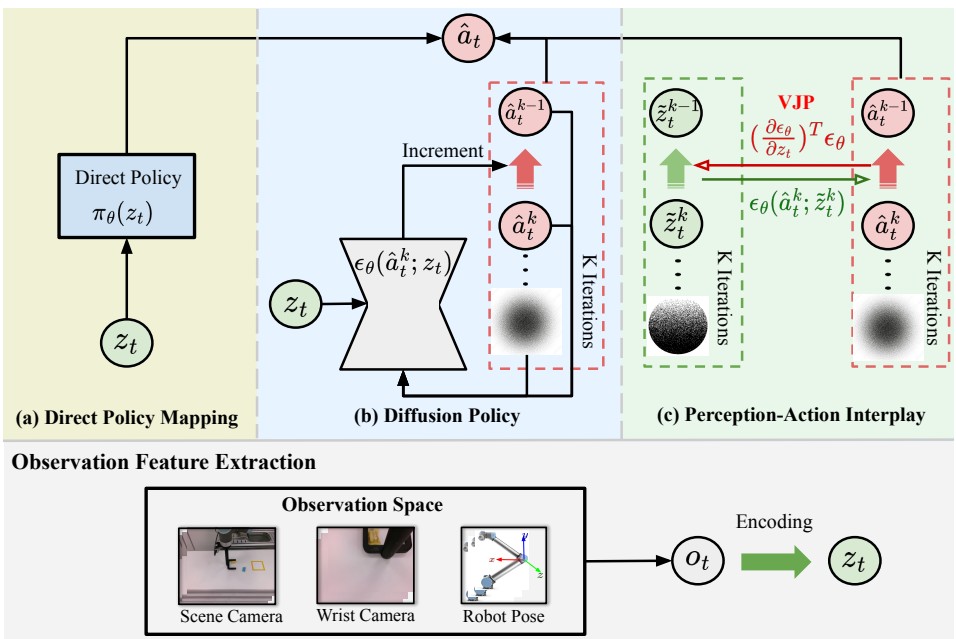

Figure 1: **Use of Observation Features.** **(a)** Conventional methods map observation features directly to actions. **(b)** DP models action distributions through incremental denoising from white noise, conditioned on observation features. **(c)** DP-AG refines observation features via noise predictions, establishing a mutually reinforcing cycle between perception and action.

because it models the continuity within each short-horizon action sequence, where intermediate action feedback can be derived from its noise predictions at each diffusion step. DP-AG first grounds observation features in a Gaussian posterior via variational inference, capturing uncertainty from observation inputs. To enable dynamic perception, we introduce an action-guided stochastic differential equation (SDE) in which latent features evolve across diffusion steps, driven by action-conditioned noise predictions. Here, the Vector–Jacobian Product (VJP) from the diffusion model acts as a structured stochastic force that shape feature evolution.

Although raw observations contain all available information, their usefulness for policy learning depends on how they are internally represented. Unlike static encoders that keep features fixed over an entire action sequence generation, DP-AG continuously refines them using action feedback. This process mirrors active perception in biological agents, where even if the external scene remains unchanged, sensory inputs are reinterpreted in the context of ongoing actions. This captures the essence of *Act to See, See to Act*: the same observation is continually reinterpreted as actions unfold, where *see* refers to the evolving latent representation refined through action feedback rather than new external sensing. As shown in Figure 1, DP-AG conditions latent evolution on action-guided noise, using the continuity of action diffusion to keep perceptual dynamics aligned with the denoising of actions. A cycle-consistent contrastive loss ensures that latent evolution remains coherent with action diffusion, preventing excessive drift and reinforcing the perception–action loop. Combined with variational inference, these components yield a principled and adaptive representation that produces smoother, more context-aware trajectories, as confirmed by both synthetic and real-robot experiments.

Theoretically, DP-AG introduces an action-guided latent SDE, derives a principled variational lower bound, and rigorously proves that the proposed cycle-consistent contrastive loss enforces continuous and coherent trajectories in both perception and action spaces. Empirically, DP-AG consistently outperforms state-of-the-art methods in success rate, convergence speed, and action smoothness, achieving gains of 6% in Push-T and 13% in Dynamic Push-T benchmarks, and delivering at least 23% higher manipulation success and approximately 60% smoother actions on real-world UR5 robot tasks compared to the baseline DP. Our contributions are summarized below:

- We propose a novel observation representation learning that establishes a closed perception–action loop by refining latent observation features via VJP-guided noise predictions derived from DP.

- We formulate an action-guided latent SDE, derive a variational lower bound, and prove that cycle-consistent InfoNCE enforces mutual smoothness in both latent and action trajectories.
- We validate the effectiveness of DP-AG in both simulation and real-world scenarios, demonstrating consistent and significant improvements in task success rate, convergence speed, and smoothness of generated actions compared to state-of-the-art methods.

## 2 Related Work

### 2.1 Imitation Learning

IL provides an alternative to reinforcement learning (RL) by removing the need for explicit reward signals [Sutton et al., 1999, Osa et al., 2018]. Existing methods include Behavioral Cloning (BC) [Bain and Sammut, 1995, Mandlekar et al., 2022, Florence et al., 2022], which learns the mapping from observations to actions, and Inverse RL [Arora and Doshi, 2021] that infers implicit rewards from expert demonstrations. Recent work [Kim et al., 2024, Black et al., 2024] extends BC by incorporating VLMs into the perception pipeline, pushing robotic manipulation toward near-human capabilities. However, existing methods decouple perception and action: the observation feature is frozen during each action sequence generation, without adapting to the evolving action refinements. This constraint often leads to abrupt or discontinuous motions. In contrast, we propose a perception–action interplay where latent continuity and action smoothness reinforce each other, yielding more coherent behavior.

### 2.2 Generative Models in Policy Learning

Generative models have shown promise in modeling continuous action trajectories. DP [Chi et al., 2023] leverages Denoising Diffusion Probabilistic Models (DDPMs) [Ho et al., 2020] to iteratively refine action predictions using diffusion models. Policy-guided diffusion [Jackson et al., 2024] generates synthetic trajectories aligned with a target policy for offline RL, while Diffusion-QL [Wang et al., 2023] uses diffusion models to learn expressive Q-functions. Recently, flow-matching methods [Lipman et al., 2023, Hu et al., 2024, Zhang et al., 2025] model actions as deterministic flow ordinary differential equations (ODEs), achieving faster inference without comprising generation quality. Building on these, DP-AG extends action-space continuity of DP into the latent observation space, forming a dynamic perception-action loop that further enhances action smoothness.

### 2.3 Latent Observation Modeling in Agent Control

Modeling latent observations for control has been explored through planning and generative methods. PlaNet [Hafner et al., 2019] and Dreamer [Wu et al., 2023] use variational autoencoders (VAEs) [Kingma et al., 2014] to simulate future states for RL. Embed to Control [Watter et al., 2015] uses VAEs to learn latent states from images for control policy design. In contrast, DP-AG closes the perception-action loop by reparameterizing latent observations with VJP-guided noise from DP, extending variational inference to capture action-guided evolution.

## 3 Preliminaries

In imitation learning, the goal is to learn a policy $\pi_\theta(a_t|o_t)$ that replicates expert behavior, where $o_t$ and $a_t$ denote the observation and action at a static time point $t$. An encoder $f_\psi$ extracts static features $z_t = f_\psi(o_t)$, which are held fixed while the policy generates the corresponding $a_t$, meaning there is no feedback from action generation to perceptual features within the same short-horizon. Given an expert demonstration dataset $\mathcal{D} = \{(o_t, a_t)\}$, standard BC minimizes the supervised loss:

$$\mathcal{L}_{\text{BC}}(\theta, \psi) = \mathbb{E}_{(o_t,a_t)\sim\mathcal{D}} \left[ \|\pi_\theta(f_\psi(o_t)) - a_t\|_2^2 \right]. \tag{1}$$

Rather than the direct mapping, DP models the conditional distribution $p_\theta(a_t|z_t)$ as a denoising diffusion process. Starting from white noise $\epsilon \sim \mathcal{N}(0, I)$, actions are generated over $K$ denoising steps using a learned noise predictor $\epsilon_\theta$, conditioned on observation features.

$$\hat{a}_t = \hat{a}_t^0 = \hat{a}_t^K - \sum_{k=1}^{K} g(k) \cdot \epsilon_\theta(\hat{a}_t^k, z_t, k), \tag{2}$$

where $a_t^K$ is the white noise, $g(k)$ is a noise schedule, and $\epsilon_\theta$ predicts the noise at diffusion step $k$. The diffusion model is trained by minimizing the noise matching objective [Ho et al., 2020]:

$$\mathcal{L}_{\text{DP}}(\theta, \psi) = \mathbb{E}_{(o_t,a_t)\sim\mathcal{D}, k\sim\mathcal{U}(1,K)} \left[ \|\epsilon_\theta(a_t^k, f_\psi(o_t), k) - \epsilon\|_2^2 \right]. \tag{3}$$

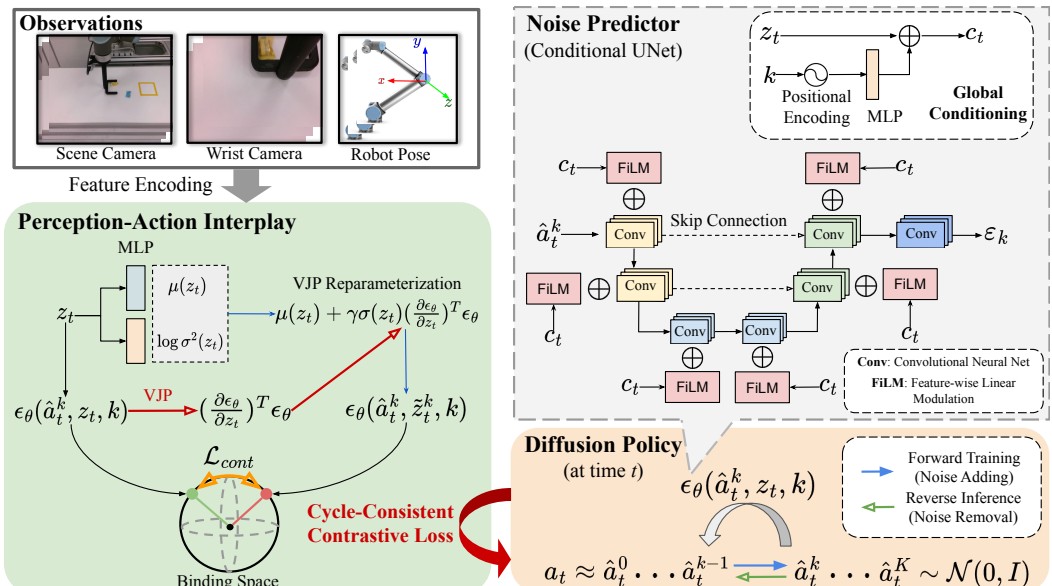

Figure 2: **Method Overview**. While Diffusion Policy (DP) generates actions from static observation features, our DP-AG establishes a dynamic perception–action loop by guiding feature evolution via the VJP of DP's predicted noise. To reinforce interplay, a cycle-consistent contrastive loss aligns noise predictions from static and evolving features, enabling mutual perception–action influence.

Through the progressive refinement of noisy actions, DP estimates the continuity underlying discrete expert actions, which allows the policy to generate smooth transitions based on discrete observations.

To leverage this continuity for perception, we introduce DP-AG, which extends DP by propagating action refinement into the latent observations through Vector–Jacobian Products (VJPs) of noise predictions, enabling features to evolve dynamically alongside actions throughout diffusion. Although we instantiate DP-AG with DDPMs, the method generalizes to other generative families such as score-based diffusion [Song et al., 2021] and flow matching [Lipman et al., 2023], provided that intermediate action refinements are available (see Appendix A for extending DP-AG to flow matching).

## 4   Our Approach

We propose Action-Guided Diffusion Policy (DP-AG), a latent generative model that explicitly captures the dynamic interplay between perception and action. As illustrated in Figure 2, DP-AG extends static variational inference by introducing an SDE over observation features, where latent representations evolve under action-guided VJPs of the diffusion noise predictions rather than being driven by static white noise. This replaces random perturbations with structured stochastic forces that align uncertainty in the latent dynamics with the task's perception–action structure. To further reinforce this interplay, we introduce a cycle-consistent contrastive loss that enforces agreement between action noise predictions conditioned on static and evolving latents, thereby closing the perception–action loop and enabling coherent updates throughout the diffusion process. Note that, this interplay is modeled within each action sequence generation $a_t$, occurring between time points $t$ and $t+1$ across the $K$ diffusion steps that connect them.

### 4.1   Latent Observation Modeling via Variational Inference

In real-world settings, observations are high-dimensional and noisy, which makes it impractical to rely on deterministic representations. To model this inherent uncertainty, we begin by constructing a variational posterior over observation features:

$$q_\phi(z_t|o_t) = \mathcal{N}(\mu_\phi(z_t), \sigma_\phi^2(z_t)), \tag{4}$$

where $\mu_\phi(z_t)$ and $\sigma_\phi^2(z_t)$ capture the mean and variance of observation features. Sampling latent observations via the reparameterization trick [Kingma et al., 2015]:

$$z_t = \mu_\phi(z_t) + \sigma_\phi(z_t) \odot \epsilon; \quad \epsilon \sim \mathcal{N}(0, I), \tag{5}$$

provides a stochastic encoding of these features, which ensures that the policy can reason about uncertainty in sensory inputs. However, it still treats perception as static: once extracted, observation features remain fixed during the corresponding action generation and cannot adapt to refinements.

## 4.2 Action-Guided Latent Evolution through Stochastic Differential Equations

To capture continuity from action diffusion, we model latent observation evolution using a SDE, where the drift is guided by action refinement. At each step $k$, the latent feature $z_t$ evolves as:

$$d\tilde{z}_t^k = \text{VJP}(\hat{a}_t^k, z_t)\, dt + \sigma_\phi(z_t)\, dW_t, \tag{6}$$

where the drift term $\text{VJP}(\hat{a}_t^k, z_t)$ is computed from the action noise prediction:

$$\text{VJP}(\hat{a}_t^k, z_t) = \left( \frac{\partial \epsilon_\theta(\hat{a}_t^k, z_t, k)}{\partial z_t} \right)^\top \epsilon_\theta(\hat{a}_t^k, z_t, k). \tag{7}$$

This follows stochastic adjoint sensitivity methods [Li et al., 2019], which show that VJP-based updates efficiently propagate gradients through SDEs without computing full Jacobians; modern autodiff frameworks make these computations practical with negligible overhead.

Beyond efficiency, the VJP serves as a task-driven attentional force, guiding latent features toward the parts of the observation that most effectively reduce uncertainty in predicting the next-step action noise. Because its drift is computed from the DP's instantaneous noise prediction, latent evolution is phase-aligned with action diffusion: moving forward during noise addition and reversing during action denoising. Unlike stochastic adjoint methods that separate forward evolution from backward updates, leaving latent features fixed during action inference, DP-AG's VJP delivers immediate feedback-driven updates at every diffusion step. This mechanism preserves the full Jacobian structure in real time, keeps latent perception features phase-aligned with ongoing action diffusion, and captures the instantaneous sensitivity between actions and observation features.

Thus, instead of a static variational posterior, we reparameterize with an action-guided noise term:

$$\tilde{z}_t^k = \mu_\phi(z_t) + \gamma \sigma_\phi(z_t) \odot \text{VJP}(\hat{a}_t^k, z_t), \tag{8}$$

where $\gamma$ controls VJP strength (see ablation studies in Appendix H.4 for its effect). This formulation allows the latent observations to evolve with the action refinements; the variational lower bound for the corresponding optimization is detailed in Section 5.1.

## 4.3 Cycle-Consistent Contrastive Learning

While VJP-guided SDEs enable dynamic latent evolution, they do not guarantee coherent alignment with the action diffusion, which is significant for achieving perception–action interplay. To bridge this gap, we introduce a cycle-consistent contrastive loss that aligns noise predictions with evolving latent dynamics at each diffusion step $k$:

- $\varepsilon_k = \epsilon_\theta(\hat{a}_t^k, z_t, k)$ is the noise prediction conditioned on the static latent $z_t$ at the step $k$;
- $\tilde{\varepsilon}_k = \epsilon_\theta(\hat{a}_t^k, \tilde{z}_t^k, k)$ is the noise prediction conditioned on the VJP-guided latent $\tilde{z}_t^k$ at the step $k$.

An InfoNCE [Chen et al., 2020] loss promotes the similarity of matched pairs while pushing apart mismatched ones within the same mini-batch (see Appendix H.6 for temperature parameter tuning):

$$\mathcal{L}_{\text{cont}} = -\frac{1}{B} \sum_{i=1}^{B} \log \frac{\exp\left( \text{sim}\left( \varepsilon_k^i, \tilde{\varepsilon}_k^i \right) / \tau \right)}{\sum_{j \neq i} \exp\left( \text{sim}\left( \varepsilon_k^i, \tilde{\varepsilon}_k^j \right) / \tau \right)}, \tag{9}$$

where $\text{sim}(u, v) = \frac{u^\top v}{||u|| ||v||}$ is cosine similarity, with $\tau$ as the temperature and $B$ the mini-batch size.

Our contrastive loss enforces cycle consistency between perception and action: noise predictions conditioned on static latents guide continuous latent updates via VJP, and the refined latents improve subsequent noise predictions. Unlike MSE-based objectives that rigidly match features, the contrastive loss promotes a clustering effect: it pulls evolved features toward their static counterparts to preserve semantic grounding, while pushing them away from features of other observations to maintain task-relevant distinctions. This bounded adaptation keeps updates semantically meaningful, enabling smooth refinement without arbitrary drift and reinforcing coherence in both latent and action trajectories (see Section 5.2 for theory and Section 6.1 for empirical validation).

### 4.4 Training Objective

The overall training objective combines three components: noise matching from DPs, variational optimization for latent evolution, and contrastive loss for the perception-action interplay:

$$\mathcal{L}_{\text{DP-AG}} = \mathcal{L}_{\text{DP}} + \lambda_{\text{cont}}\mathcal{L}_{\text{cont}} + \lambda_{\text{KL}}\mathcal{L}_{\text{KL}}. \tag{10}$$

Jointly optimizing these terms enables DP-AG to co-evolve perception and action, resulting in more adaptive policies (see Appendix H.5 for ablation studies on hyperparameter tuning).

### 4.5 Intuitions Behind DP-AG

We now explain why DP-AG works in practice. A static encoder yields a one-shot representation of the observation that stays fixed during action generation, forcing the policy to commit to an interpretation before actions unfold. DP-AG instead refines the latent representation step by step, guided by feedback from the evolving action sequence. Think of driving a car: the view through the windshield may remain unchanged, yet the driver's focus shifts as they act, paying more attention to the curve's edge while turning, or to nearby cars when accelerating.

The VJP supplies both the *direction* and *magnitude* of this latent drift: it points features toward the adjustments that most reduce action uncertainty, with larger updates when the model is less confident and smaller ones when it is certain. Meanwhile, the cycle-consistent contrastive loss anchors these adaptations, keeping updated features aligned with their static counterparts and preventing excessive drift. Together, this **action-driven refinement** and **bounded consistency** yield smoother, more stable, and context-aware trajectories, as confirmed in our experiments.

## 5 Theoretical Insights

### 5.1 Action-Guided Variational Lower Bound

To learn a dynamic latent observation that evolves with the action diffusion, we define a posterior $q_\phi(\tilde{z}_t^k|z_t, \hat{a}_t^k)$ conditioned on both the static latent $z_t$ and the denoised action $\hat{a}_t^k$ at step $k$.

**Variational Objective.** The marginal likelihood of the observed noise throughout the diffusion is:

$$\log p(\varepsilon_k|z_t) = \log \int p(\tilde{\varepsilon}_k|\tilde{z}_t^k)\, p(\tilde{z}_t^k|z_t)\, d\tilde{z}_t^k. \tag{11}$$

Applying Jensen's inequality with the posterior, we obtain the ELBO:

$$\log p(\varepsilon_k|z_t) \geq \mathbb{E}_{q_\phi(\tilde{z}_t^k|z_t, \hat{a}_t^k)}\left[\log p(\tilde{\varepsilon}_k|\tilde{z}_t^k)\right] - \text{KL}\left(q_\phi(\tilde{z}_t^k|z_t, \hat{a}_t^k) \,\|\, p(\tilde{z}_t^k|z_t)\right). \tag{12}$$

Maximizing the ELBO encourages $\tilde{z}_t^k$ to capture action-driven updates while staying close to the prior conditioned on $z_t$. Full derivation of the ELBO is provided in Appendix B.

**Likelihood Term.** The likelihood term $p(\tilde{\varepsilon}_k|\tilde{z}_t^k)$ models the distribution of action noise conditioned on the evolving latent $\tilde{z}_t^k$, and can be optimized using a noise matching loss similar to DP:

$$\mathcal{L}_{\text{LH}} = \mathbb{E}_{(o_t, a_t)\sim\mathcal{D}, k\sim\mathcal{U}(1,K)}\left[\|\epsilon_\theta(\hat{a}_t^k, \tilde{z}_t^k, k) - \epsilon\|_2^2\right]. \tag{13}$$

This likelihood term encourages absolute accuracy in noise prediction but overlaps with the contrastive loss that emphasizes relative similarity (see the detailed ablation in Appendix H.3). To avoid redundancy and instability, we use only the contrastive loss.

**Kullback–Leibler Divergence Term.** The variational posterior is modeled as:

$$q_\phi(\tilde{z}_t^k|z_t, \hat{a}_t^k) = \mathcal{N}(\mu_\phi(z_t, \hat{a}_t^k), \sigma_\phi^2(z_t, \hat{a}_t^k)), \tag{14}$$

while the conditional prior can be modeled as a Gaussian distribution:

$$p(\tilde{z}_t^k|z_t) = \mathcal{N}(z_t, I), \tag{15}$$

which encourages the evolved latent to stay near the static latent initially encoded from the observation. The Kullback–Leibler (KL) divergence thus becomes (derivation of Equation 16 is in Appendix C):

$$\text{KL}(q_\phi(\tilde{z}_t^k|z_t, \hat{a}_t^k) \,\|\, p(\tilde{z}_t^k|z_t)) = \frac{1}{2}\sum_{i=1}^{d}\left(\sigma_{\phi,i}^2 + (\mu_{\phi,i} - z_{t,i})^2 - 1 - \log\sigma_{\phi,i}^2\right), \tag{16}$$

where $d$ is the latent dimension, and $\mu_\phi(\cdot)$ and $\log\sigma_\phi^2(\cdot)$ are parameterized by a separate linear layer.

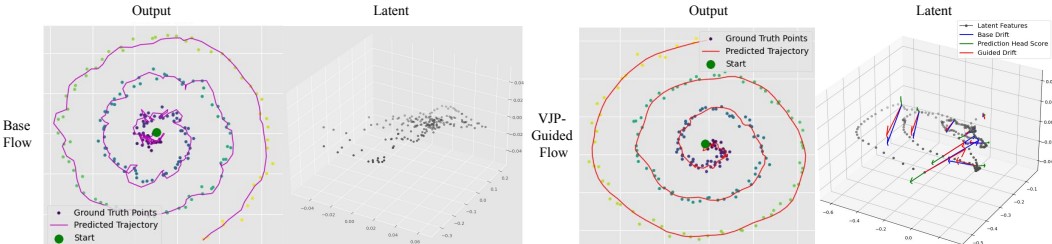

Figure 3: **Regression results on irregular spirals. Left**: Trajectories and latent dynamics predicted by the Base Flow. **Right**: The VJP-Guided Flow continuously refines latents through output-guided corrections, which results in smoother and more coherent trajectories in both output and latent spaces.

## 5.2 Contrastive Similarity Promotes Latent-Action Continuity

In this section, we theoretically demonstrate that minimizing our cycle-consistent InfoNCE loss leads to smooth and coherent transitions in both latent and action spaces. The key insight is that minimizing the contrastive loss enforces strong similarity between static and dynamic noise predictions; under Lipschitz continuity, this similarity tightly bounds the drift in latent dynamics.

**Noise Similarity Lower Bound.** We first show that minimizing the contrastive loss imposes a lower bound on the similarity between noise predictions $\varepsilon_k^i$ and $\tilde{\varepsilon}_k^i$. Proof of Lemma 1 is in Appendix D.

**Lemma 1** (Noise Similarity Lower Bound). *For unit-normalized vectors $\varepsilon_k^i$ and $\tilde{\varepsilon}_k^i$, and a temperature $\tau > 0$, if the InfoNCE loss satisfies $\mathcal{L}_{cont} \leq \alpha$ for some small constant $\alpha$, then for each $i \in \{1, \ldots, B\}$, the similarity between corresponding pairs is bounded accordingly:*

$$\underbrace{\text{sim}\left(\varepsilon_k^i, \tilde{\varepsilon}_k^i\right)}_{positive\ pair\ similarity} \geq \tau(\ln(B-1) - \alpha) - 1. \tag{17}$$

**Continuity Induced by Noise Alignment.** We then demonstrate that the lower bound on the noise similarity provides an upper bound on the latent drift, meaning that the contrastive alignment also leads to smooth transitions in the latent space. Proof of Theorem 1 is in Appendix E.

**Theorem 1** (Continuity Upper Bound). *Suppose $\epsilon_\theta(\cdot)$ is L-Lipschitz with respect to z, and that $\tilde{z}_t^{k+1}$ evolves via the VJP of $\epsilon_\theta$. Then under the contrastive constraint $\mathcal{L}_{cont} \leq \alpha$, we have:*

$$\|\tilde{z}_t^{k+1} - \tilde{z}_t^k\|_2^2 \leq L^2 \|\epsilon_\theta(\hat{a}_t^k, z_t, k) - \epsilon_\theta(\hat{a}_t^k, \tilde{z}_t^k, k)\|_2^2 \leq 2L^2(2 - \tau\ln(B-1) + \tau\alpha). \tag{18}$$

Together, Lemma 1 and Theorem 1 establish a theoretical connection between minimizing our cycle-consistent contrastive loss and promoting temporal continuity in both latent and action trajectories.

## 6 Empirical Evaluations

### 6.1 Latent and Action Continuity on an Irregular Time-Series Regression

To empirically validate DP-AG's ability to improve latent and action continuities, we conducted experiments on the irregular spiral dataset, originally introduced in the neural ODE evaluations [Chen et al., 2018]. This dataset is designed to evaluate continuous-time dynamics under irregular sampling, making it well-suited for evaluating the impact of the VJP-guided evolution on latent dynamics.

**Dataset.** Following [Chen et al., 2018], the dataset contains 1,000 samples of 2D spiral trajectories, with half following a clockwise pattern and the other half counterclockwise, each sampled at 100 irregular time points. The spiral exhibits continuous radial growth with Gaussian noise ($\sigma \in [0.02, 0.1]$) added to simulate real-world variability with randomly assigned angular velocities.

**Method.** We compare two models:

- **Base Flow:** An LSTM with 128 hidden units and 2 layers is used to extract discrete latent features $z_t$, followed by a Multi-Layer Perceptron (MLP) to predict the regression output $y_\theta(z_t)$.
- **VJP-Guided Flow:** Augmenting the base flow with a VJP-guided SDE, where the latent feature $\tilde{z}_t$ evolves according to: $d\tilde{z}_t = \left(\frac{\partial y_\theta(z_t)}{\partial z_t}\right)^\top y_\theta(z_t)\, dt + \sigma_\phi(z_t)\, dW_t$ with the base drift $\mu_\phi(z_t)$ and base log-variance $\log \sigma_\phi^2(z_t)$ are each predicted by a separate linear layer.

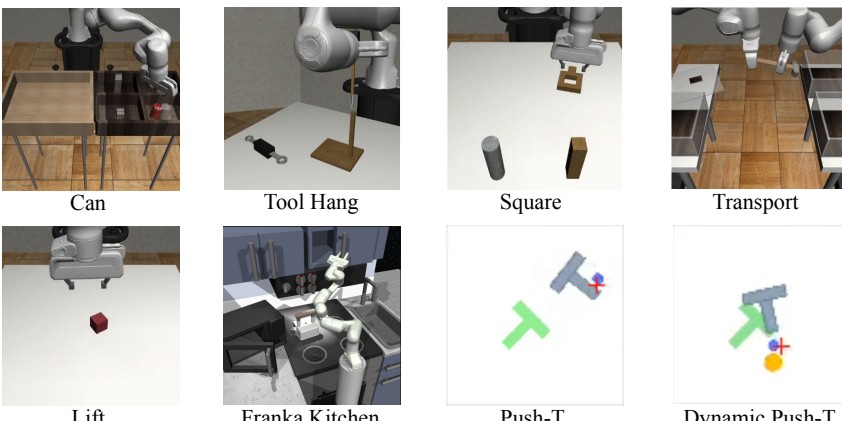

Figure 4: Benchmark simulation environments include Robomimic, Franka Kitchen, Push-T, and Dynamic Push-T, with task and dataset details provided in Appendix F.

Table 1: **Target coverage score on Push-T and Dynamic Push-T tasks. img** and **kp** refer to the observation modalities: RGB images or 2D keypoints.

| Type | Method | Push-T | | Dynamic Push-T |
|------|--------|--------|--------|----------------|
| | | img | kp | img |
| Mapping | LSTM-GMM [Mandlekar, 2022] | 0.69±0.02 | 0.67±0.03 | 0.34±1.24 |
| | IBC [Florence, 2022] | 0.75±0.02 | 0.90±0.02 | 0.52±0.98 |
| | BET [Shafiullah, 2022] | 0.80±0.02 | 0.79 ±0.02 | 0.58±1.35 |
| Flow | FlowPolicy [Zhang, 2025] | 0.85±0.01 | 0.88±0.01 | 0.53±0.88 |
| | AdaFlow [Hu, 2024] | 0.87±0.02 | 0.91±0.01 | 0.67±0.79 |
| Diffusion | DP [Chi, 2023] | 0.87±0.04 | 0.95±0.03 | 0.65±0.85 |
| | DP-AG (ours) | **0.93**±0.02 | **0.99**±0.01 | **0.80**±0.53 |

Both models are trained for 100 epochs with Adam optimizer (learning rate $10^{-3}$ and batch size 32).

**Results.** As shown in Figure 3, VJP-Guided Flow generates smoother and more coherent trajectories than the Base Flow, reducing the MSE from 0.0095 to 0.0052 (a 45.3% improvement). Importantly, latent visualizations show that while the Base Flow yields scattered latent states, VJP guidance shapes a structured latent manifold aligned with output predictions. This illustrates our perception-action interplay: predictions provide gradient feedback that dynamically refines latent embeddings through the VJP. As a result, latent embeddings and predictions evolve in synchrony, enhancing continuity in both representation and regression, which is consistent with our theoretical findings in Theorem 1.

### 6.2 Experiments on Simulation Benchmarks

We then evaluate our DP-AG on the simulation benchmarks, illustrated in Figure 4.

**Robomimic.** Robomimic [Mandlekar et al., 2022] is a large-scale benchmark for robotic manipulation consisting of five tasks with nine datasets: Can (ph/mh), Square (ph/mh), Transport (ph/mh), Tool Hang (ph), and Lift (ph/mh). Here, **ph** denotes proficient human demonstrations, and **mh** indicates a mix of proficient and non-proficient demonstrations.

**Franka Kitchen.** Franka Kitchen [Gupta et al., 2020] is a simulation benchmark with a 9-DoF Franka arm performing four household tasks (**t1**, **t2**, **t3**, and **t4**) per trajectory, using 566 human demonstrations across 7 interactive objects. Available through platforms like Gymnasium-Robotics [1].

**Push-T.** Push-T [Chi et al., 2023] is a manipulation task adapted from IBC [Florence et al., 2022], where a circular end-effector pushes a T-shaped block to a target location. Both the block and end-effector start at random positions. Observations consist of either RGB images (**img**) or a set of nine 2D keypoints (**kp**) outlining the T block's shape, along with the position of the end-effector.

**Dynamic Push-T (Ours).** Many IL benchmarks are nearly saturated: they are deterministic, scripted, and lack diversity, often solvable without closed-loop feedback [Jia et al., 2024]. To evaluate real-time

---

[1] https://robotics.farama.org/envs/franka_kitchen/franka_kitchen/

Table 2: **Success rates across Robomimic and Franka Kitchen tasks. ph**: proficient human demos; **mh**: mixed-quality demos; **t1** to **t4** denote task IDs in Franka Kitchen.

| Type | Method | Lift | | Can | | Square | | Transport | | ToolHang | Franka Kitchen | | | |
|---|---|---|---|---|---|---|---|---|---|---|---|---|---|---|
| | | ph | mh | ph | mh | ph | mh | ph | mh | ph | t1 | t2 | t3 | t4 |
| Mapping | LSTM-GMM [Mandlekar, 2022] | **1.00** | **1.00** | **1.00** | 0.98 | 0.82 | 0.64 | 0.88 | 0.44 | 0.68 | 1.00 | 0.90 | 0.74 | 0.34 |
| | IBC [Florence, 2022] | 0.94 | 0.39 | 0.08 | 0.00 | 0.03 | 0.00 | 0.00 | 0.00 | 0.00 | 0.99 | 0.87 | 0.61 | 0.24 |
| | BET [Shafiullah, 2022] | **1.00** | **1.00** | **1.00** | **1.00** | 0.76 | 0.68 | 0.38 | 0.21 | 0.58 | 0.99 | 0.93 | 0.71 | 0.44 |
| Flow | FlowPolicy [Zhang, 2025] | 0.98 | 0.95 | 0.98 | 0.98 | 0.86 | 0.90 | 0.88 | 0.82 | 0.85 | 0.96 | 0.86 | 0.95 | 0.87 |
| | AdaFlow [Hu, 2024] | **1.00** | **1.00** | **1.00** | 0.96 | 0.98 | 0.96 | 0.92 | 0.80 | 0.88 | 0.99 | 0.89 | 0.92 | 0.83 |
| Diffusion | DP [Chi, 2023] | **1.00** | **1.00** | **1.00** | **1.00** | 0.98 | 0.98 | **1.00** | 0.89 | 0.95 | **1.00** | **1.00** | **1.00** | 0.99 |
| | DP-AG (ours) | **1.00** | **1.00** | **1.00** | **1.00** | **1.00** | **1.00** | **1.00** | **0.94** | **0.98** | **1.00** | **1.00** | **1.00** | **1.00** |

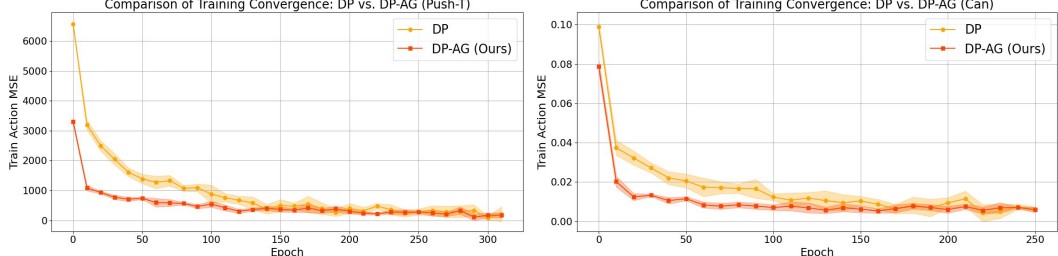

Figure 5: **Convergence Plots.** Training action MSE over epochs on Push-T and Robomimic Can.

adaptability, we propose *Dynamic Push-T*, which augments Push-T with a moving ball that bounces unpredictably and intermittently interferes with manipulation. Since the ball's trajectory varies in each episode, the agent must adapt online, blocking or avoiding the ball while pursuing the main objective. This develops a dynamic and unscripted challenge that evaluates true adaptability beyond replaying demonstrations. Details of this benchmark are provided in Appendix F.

**Baselines and Evaluation Metrics.** We compare DP-AG against IL baselines that include direct mapping (LSTM-GMM [Mandlekar et al., 2022], IBC [Florence et al., 2022], BET [Shafiullah et al., 2022]), diffusion-based (DP [Chi et al., 2023]), and flow-matching methods (AdaFlow [Hu et al., 2024], FlowPolicy [Zhang et al., 2025]). Results are from either our reimplementation or the original papers, averaged over 5 training seeds and multiple evaluation seeds (50 for Push-T/Kitchen, 22 for Robomimic, 50 for Dynamic Push-T). We use success rate as the main metric, with target area coverage for Push-T variants. Additional implementation details are in Appendix G.

**Results.** Tables 1 and 2 show that DP-AG consistently outperforms baselines across Robomimic, Franka Kitchen, Push-T, and Dynamic Push-T, achieving near-perfect success on static tasks and the highest coverage on dynamic ones. These gains come from enforcing smooth latent and action trajectories, which are especially important in Dynamic Push-T. DP-AG also converges faster than DP (Figure 5 with additional plots in Appendix I). Ablation results are provided in Appendix H, and computational analysis is detailed in Appendix H.1.

### 6.3 Real-World Evaluation on UR5 Robot Arm

To evaluate the real-world performance of our DP-AG, we deploy DP and DP-AG policies onto a UR5 robotic arm in three visuomotor manipulation tasks: **Painting**, **Candy Push**, and **Peg-In-Hole Insertion** (Figure 6). These tasks are designed to evaluate different aspects of perception–action interplay and the policy's ability to generalize from partially observable inputs. The details (dataset, training, etc.) of the real-world manipulation experiments are provided in Appendix J.

**Painting (Planar Precision).** The robot is tasked with tracing heart-shaped and circular-shaped paths using a paintbrush. This task requires smooth and continuous motion to avoid over-painting or streaking. Our DP-AG improves both path fidelity and smoothness compared to DP.

**Candy Push (Object-aware Adaptation).** The end-effector pushes small candies into a designated goal area. Object positions are randomized across trials. DP-AG's latent evolution allows it to adapt to variations in candy layouts and locations, which results in fewer collisions and smoother trajectories.

**Peg-in-Hole (3D Visual Reasoning).** The robot must insert a circular peg into a vertical hole using only RGB inputs from scene and wrist cameras, without explicit depth sensing, requiring the policy

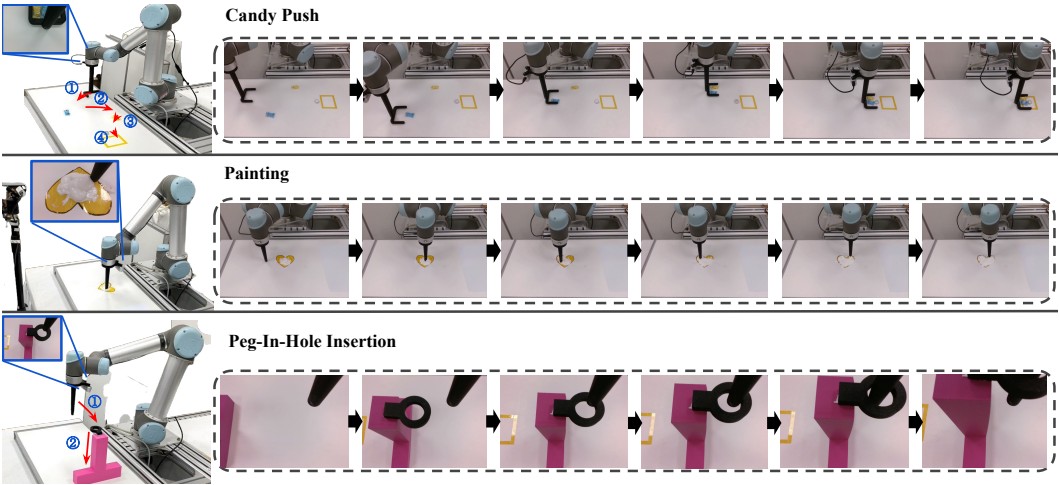

Figure 6: Real-world evaluation on a UR5 robot arm across three manipulation tasks.

Table 3: **Performance on Real-World UR5 Tasks.** Mean and standard deviation are reported.

| Task | Method | Success Rate (%) | Smoothness (Avg. Jerk) | IoU (%) | Time to Complete (s) |
|---|---|---|---|---|---|
| Painting | DP | – | $0.083 \pm 0.014$ | $68.9 \pm 5.2$ | $49.5 \pm 4.1$ |
| | DP-AG | – | $\mathbf{0.032 \pm 0.009}$ | $\mathbf{92.1 \pm 3.4}$ | $\mathbf{18.0 \pm 3.2}$ |
| Candy Push | DP | $65.0 \pm 8.4$ | $0.107 \pm 0.016$ | – | $24.0 \pm 3.9$ |
| | DP-AG | $\mathbf{90.0 \pm 5.5}$ | $\mathbf{0.039 \pm 0.011}$ | – | $\mathbf{9.5 \pm 2.6}$ |
| Peg-in-Hole | DP | $0.0 \pm 0.0$ | $0.096 \pm 0.017$ | – | – |
| | DP-AG | $\mathbf{85.0 \pm 6.0}$ | $\mathbf{0.036 \pm 0.008}$ | – | $\mathbf{13.0 \pm 2.1}$ |

to infer 3D geometry from indirect cues. Baseline DP offers no mechanism to adapt when the hole is slightly misaligned. Upon contact, it repeatedly executes the same ineffective motion, failing in all trials. In contrast, DP-AG leverages action feedback: each blocked step triggers a VJP-guided latent update that sharpens sensitivity to contact-region cues (*e.g.,* rim alignment). Over a few refinements, the same RGB inputs are reinterpreted to expose geometry cues sufficient for correction, enabling successful insertions under occlusion and unseen perturbations.

**Evaluation.** We evaluate our DP-AG against DP on the three tasks, each repeated over 20 trials. For Painting, we report IoU between the painted and target shapes; for Candy Push and Peg-In-Hole, we report success rate. Moreover, we measure trajectory smoothness via average jerk and task completion time. As shown in Table 3, DP-AG consistently outperforms DP across all metrics. Notably, the largest gain occurs in the peg-in-hole task, where DP-AG's dynamic latent updates help infer 3D information from 2D observations, which is an ability DP lacks due to its static features.

## 7 Conclusion

We introduce DP-AG, a representation learning framework that closes the perception–action loop in diffusion policies. By evolving latent observations through an action-guided SDE, DP-AG transforms diffusion noise gradients into structured perceptual updates via VJPs. A cycle-consistent contrastive loss aligns static and evolving latents, enabling continuous and bidirectional coupling between perception and action throughout diffusion. We derive a principled ELBO and prove that contrastive alignment enforces mutual continuity in latent and action trajectories. Empirically, DP-AG achieves state-of-the-art performance, especially under partial observability and dynamic conditions, demonstrating that perception–action interplay is significant for effective and adaptive policy learning.

**Broader Impacts.** DP-AG advances imitation learning by modeling dynamic perception-action interplay, enabling smoother and more context-aware robotic manipulation and reducing the risk of failure or unintended motions. This capability is particularly valuable in safety-critical domains such as automation, manufacturing, and assistive robotics, where robust decision-making is significant for effective human-robot collaboration. However, relying on expert demonstrations can introduce biases or suboptimal behaviors into the policy. To mitigate these risks, it is important to rigorously validate the quality and diversity of demonstration data to minimize unintended actions.

## Acknowledgments and Disclosure of Funding

This work was partly supported by NSERC Discovery, CFI-JELF, NSERC Alliance, Alberta Innovates and PrairiesCan grants.

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

# Appendix

## A    Extension to Flow Matching Models

While we demonstrate DP-AG built upon DDPMs (as in DP), the proposed perception-action interplay can be extended to alternative generative models that produce smooth action trajectories, such as flow matching models [Lipman et al., 2023, Black et al., 2024, Zhang et al., 2025].

**Background on Flow Matching.** Flow matching formulates action generation as solving an ordinary differential equation (ODE) that deterministically transforms a sample from a known base distribution (e.g., Gaussian) to the target data distribution. Given a continuous time variable $t \in [0, 1]$, the action $a(t)$ evolves according to:

$$\frac{da(t)}{dt} = v_\theta(a(t), z_t, t),$$
(19)

where $v_\theta$ denotes a learned velocity field conditioned on observation features $z_t$. The goal is to match the score of the intermediate marginal distribution $p_t(a)$ at time $t$ by minimizing a regression loss. Unlike diffusion models, flow matching generates continuous evolution directly without stochastic perturbations.

**Action-Guided Latent Evolution with Flow Matching.** To extend DP-AG to flow matching models, we leverage the structure of the learned velocity field $v_\theta$ to guide the latent dynamics. To be specific, at each time $t$, we define the evolution of the observation feature $\tilde{z}_t$ through a VJP-guided ODE:

$$\frac{d\tilde{z}_t}{dt} = \left( \frac{\partial v_\theta(a(t), z_t, t)}{\partial z_t} \right)^\top v_\theta(a(t), z_t, t),$$
(20)

where $v_\theta(a(t), z_t, t)$ serves as the action-conditioned driving force, and the VJP propagates this force back to update the observation feature $\tilde{z}_t$. This setup follows our DP-AG formulation for DDPMs but replaces the stochastic increments from noise predictions with deterministic updates guided by flow matching dynamics.

**Training Objective.** Similar to the DP-AG framework, we introduce a variational posterior $q_\phi(\tilde{z}_t | z_t, a(t))$ over the evolved latents and formulate a ELBO:

$$\mathcal{L}_{\text{ELBO}} = \mathbb{E}q_\phi \left[ \log p_\theta(v_\theta(a(t), \tilde{z}_t, t)) \right] - \text{KL} \left( q_\phi(\tilde{z}_t | z_t, a(t)) || p(\tilde{z}_t | z_t) \right),$$
(21)

where $p(\tilde{z}_t | z_t)$ is the base distribution, e.g., isotropic Gaussian centered at $z_t$. Moreover, the cycle-consistent contrastive loss can be formulated between the velocity fields $v_\theta(a(t), z_t, t)$ and $v_\theta(a(t), \tilde{z}_t, t)$ to promote the consistent evolution between the action and the latent trajectories.

**Discussion.** The flow matching extension has several benefits:

- Deterministic evolution improves sample efficiency compared to stochastic DDPM training.
- The continuous latent trajectories can be directly controlled via $v_\theta$, which simplifies the interpretation.
- The perception-action interplay remains: smoother latent evolution produces more coherent actions, and better action flow, in turn, improves the latent dynamics.

Thus, DP-AG extends beyond DDPMs, which provide a flexible framework for integrating with generative-model-based policy learning under both stochastic and deterministic dynamics.

## B    Derivation of DP-AG Variational Lower Bound

In this section, we show the full derivation of our ELBO using Jensen's inequality:

$$\log p(\varepsilon_k | z_t) \geq \mathbb{E}_{q_\phi(\tilde{z}_t^k | z_t, \hat{a}_t^k)} \left[ \log p(\tilde{\varepsilon}_k | \tilde{z}_t^k) \right] - \text{KL} \left( q_\phi(\tilde{z}_t^k | z_t, \hat{a}_t^k) \,\|\, p(\tilde{z}_t^k | z_t) \right),$$
(22)

where $q_\phi(\tilde{z}_t^k | z_t, \hat{a}_t^k)$ is the variational posterior approximating the true posterior.

We start from the exact marginal:

$$p(\varepsilon_k | z_t) = \int p(\tilde{\varepsilon}_k | \tilde{z}_t^k) p(\tilde{z}_t^k | z_t) \, d\tilde{z}_t^k.$$
(23)

Inserting the variational posterior $q_\phi(\tilde{z}_t^k|z_t, \hat{a}_t^k)$ via importance sampling:

$$p(\varepsilon_k|z_t) = \int q_\phi(\tilde{z}_t^k|z_t, \hat{a}_t^k) \frac{p(\tilde{\varepsilon}_k|\tilde{z}_t^k)p(\tilde{z}_t^k|z_t)}{q_\phi(\tilde{z}_t^k|z_t, \hat{a}_t^k)} \, d\tilde{z}_t^k = \mathbb{E}_{q_\phi}\left[\frac{p(\tilde{\varepsilon}_k|\tilde{z}_t^k)p(\tilde{z}_t^k|z_t)}{q_\phi(\tilde{z}_t^k|z_t, \hat{a}_t^k)}\right]. \tag{24}$$

Thus, we have:

$$\log p(\varepsilon_k|z_t) = \log \mathbb{E}_{q_\phi}\left[\frac{p(\tilde{\varepsilon}_k|\tilde{z}_t^k)p(\tilde{z}_t^k|z_t)}{q_\phi(\tilde{z}_t^k|z_t, \hat{a}_t^k)}\right]. \tag{25}$$

Since $\log(\cdot)$ is a concave function, by Jensen's inequality:

$$\log \mathbb{E}_{q_\phi}[X] \geq \mathbb{E}_{q_\phi}[\log X], \tag{26}$$

for any non-negative random variable X. Here, $X = \frac{p(\tilde{\varepsilon}_k|\tilde{z}_t^k)p(\tilde{z}_t^k|z_t)}{q_\phi(\tilde{z}_t^k|z_t, \hat{a}_t^k)}$ is clearly non-negative because all densities are non-negative.

Thus, applying Jensen's inequality to our case gives:

$$\begin{aligned}
\log \mathbb{E}_{q_\phi}\left[\frac{p(\tilde{\varepsilon}_k|\tilde{z}_t^k)p(\tilde{z}_t^k|z_t)}{q_\phi(\tilde{z}_t^k|z_t, \hat{a}_t^k)}\right] &\geq \mathbb{E}_{q_\phi}\left[\log\left(\frac{p(\tilde{\varepsilon}_k|\tilde{z}_t^k)p(\tilde{z}_t^k|z_t)}{q_\phi(\tilde{z}_t^k|z_t, \hat{a}_t^k)}\right)\right] \\
&= \mathbb{E}_{q_\phi}\left[\log p(\tilde{\varepsilon}_k|\tilde{z}_t^k) + \log p(\tilde{z}_t^k|z_t) - \log q_\phi(\tilde{z}_t^k|z_t, \hat{a}_t^k)\right] \\
&= \mathbb{E}_{q_\phi}\left[\log p(\tilde{\varepsilon}_k|\tilde{z}_t^k)\right] + \mathbb{E}_{q_\phi}\left[\log p(\tilde{z}_t^k|z_t)\right] - \mathbb{E}_{q_\phi}\left[\log q_\phi(\tilde{z}_t^k|z_t, \hat{a}_t^k)\right], \\
&= \mathbb{E}_{q_\phi}\left[\log p(\tilde{\varepsilon}_k|\tilde{z}_t^k)\right] - \mathrm{KL}\left(q_\phi(\tilde{z}_t^k|z_t, \hat{a}_t^k) \,\|\, p(\tilde{z}_t^k|z_t)\right).
\end{aligned} \tag{27}$$

Thus, we have proven:

$$\log p(\varepsilon_k|z_t) \geq \mathbb{E}_{q_\phi(\tilde{z}_t^k|z_t, \hat{a}_t^k)}\left[\log p(\tilde{\varepsilon}_k|\tilde{z}_t^k)\right] - \mathrm{KL}\left(q_\phi(\tilde{z}_t^k|z_t, \hat{a}_t^k) \,\|\, p(\tilde{z}_t^k|z_t)\right). \tag{28}$$

$\square$

## C   Derivation and Intuition for the KL Divergence Term in Equation 16

To regularize the latent evolution $\tilde{z}_t^k$ and preserve semantic consistency with the observation-encoded latent $z_t$, we introduce a KL divergence between a variational posterior and a conditional prior. This section provides a step-by-step derivation and intuition for the expression in Equation 16.

**Formulation.** The variational posterior is defined as a Gaussian distribution:

$$q_\phi(\tilde{z}_t^k|z_t, \hat{a}_t^k) = \mathcal{N}(\mu_\phi(z_t, \hat{a}_t^k), \mathrm{diag}(\sigma_\phi^2(z_t, \hat{a}_t^k))), \tag{29}$$

where both the mean and log-variance are predicted by linear layers. The conditional prior is set to:

$$p(\tilde{z}_t^k|z_t) = \mathcal{N}(z_t, I), \tag{30}$$

which enforces that the evolved latent remains close to its observation-driven anchor $z_t$ unless strongly influenced by the action.

**KL Between Diagonal Gaussians.** The KL divergence between two diagonal Gaussians $\mathcal{N}(\mu_q, \Sigma_q)$ and $\mathcal{N}(\mu_p, \Sigma_p)$ is given by:

$$\mathrm{KL}(q\|p) = \frac{1}{2}\sum_{i=1}^d \left[\frac{\sigma_{q,i}^2}{\sigma_{p,i}^2} + \frac{(\mu_{q,i} - \mu_{p,i})^2}{\sigma_{p,i}^2} - 1 - \log\frac{\sigma_{q,i}^2}{\sigma_{p,i}^2}\right], \tag{31}$$

where $d$ is the dimensionality of samples drawn from Guassians. Substituting $\mu_q = \mu_\phi(z_t, \hat{a}_t^k)$, $\sigma_q = \sigma_\phi(z_t, \hat{a}_t^k)$, $\mu_p = z_t$, and $\Sigma_p = I$, we obtain:

$$\mathrm{KL}(q_\phi(\tilde{z}_t^k|z_t, \hat{a}_t^k) \,\|\, p(\tilde{z}_t^k|z_t)) = \frac{1}{2}\sum_{i=1}^d \left(\sigma_{\phi,i}^2 + (\mu_{\phi,i} - z_{t,i})^2 - 1 - \log\sigma_{\phi,i}^2\right), \tag{32}$$

where $d$ is now the latent dimension, and all quantities are computed element-wise.

**Intuitions.** Each term in the KL expression serves a distinct regularization role:

- $(\mu_{\phi,i} - z_{t,i})^2$: penalizes deviation of the evolved latent mean from the observation-encoded static latent.
- $\sigma_{\phi,i}^2$: penalizes over-dispersion and encourages confident latent representations.
- $\log \sigma_{\phi,i}^2$: rewards expressive uncertainty when needed, balancing the previous term.

This KL divergence term acts as a geometric constraint to preserve alignment with the observation-anchored latent space, while still allowing dynamic evolution based on the action. Combined with our contrastive loss (Equation 9), it stabilizes training by maintaining local continuity and global discriminability.

# D  Proof of Lemma 1

Following the theoretical work on the InfoNCE loss [Parulekar et al., 2023, Wang et al., 2024], we derive the lower bound for the positive key alignment:

**Lemma** (Noise Similarity Lower Bound). *For unit-normalized vectors $\varepsilon_k^i$ and $\tilde{\varepsilon}_k^i$, and a temperature $\tau > 0$, if the InfoNCE loss satisfies $\mathcal{L}_{cont} \leq \alpha$ for some small constant $\alpha$, then for each $i \in \{1, \ldots, B\}$, the similarity between corresponding pairs is bounded accordingly:*

$$\underbrace{\text{sim}\left(\varepsilon_k^i, \tilde{\varepsilon}_k^i\right)}_{\text{positive pair similarity}} \geq \tau(\ln(B-1) - \alpha) - 1. \tag{33}$$

*Proof.* Recall that the InfoNCE loss is defined as:

$$\mathcal{L}_{\text{cont}} = -\frac{1}{B} \sum_{i=1}^{B} \log \frac{\exp\left(\text{sim}\left(\varepsilon_k^i, \tilde{\varepsilon}_k^i\right)/\tau\right)}{\sum_{j \neq i} \exp\left(\text{sim}\left(\varepsilon_k^i, \tilde{\varepsilon}_k^j\right)/\tau\right)}. \tag{34}$$

To simplify the notations, let $s_{ii} = \text{sim}(\varepsilon_k^i, \tilde{\varepsilon}_k^i)$ and $s_{ij} = \text{sim}(\varepsilon_k^i, \tilde{\varepsilon}_k^j)$ for $j \neq i$. We define:

$$p_i = \frac{\exp(s_{ii}/\tau)}{\sum_{j \neq i} \exp(s_{ij}/\tau)}. \tag{35}$$

The loss constraint $\mathcal{L}_{\text{cont}} \leq \alpha$ implies:

$$-\frac{1}{B} \sum_{i=1}^{B} \log p_i \leq \alpha \implies \sum_{i=1}^{B} \log p_i \geq -B\alpha. \tag{36}$$

Taking exponential at both sides, we have:

$$\prod_{i=1}^{B} p_i \geq \exp(-B\alpha). \tag{37}$$

To derive a lower bound on $s_{ii}$, we need the denominator $\sum_{j \neq i} \exp(s_{ij}/\tau)$ to be as small as possible. Since $s_{ij} \geq -1$ (unit vectors), the denominator is minimized when all $s_{ij} = -1$. Thus, we have:

$$\sum_{j \neq i} \exp(s_{ij}/\tau) \geq (B-1)\exp(-1/\tau), \tag{38}$$

and

$$p_i \leq \frac{\exp(s_{ii}/\tau)}{(B-1)\exp(-1/\tau)} = \frac{1}{B-1} \exp\left(\frac{s_{ii}+1}{\tau}\right), \tag{39}$$

Assume that $p_i$ is equal across all $i$ to maximize the product under the constraint by Arithmetic Mean-Geometric Mean (AM-GM) inequality [Tan and Xie, 2020]:

$$p_i = \left(\prod_{i=1}^{B} p_i\right)^{1/B} \geq \exp(-B\alpha)^{1/B} = \exp(-\alpha). \tag{40}$$

Substituting into Inequality 39, we have:

$$\exp(-\alpha) \le p_i \le \frac{1}{B-1} \exp\left(\frac{s_{ii}+1}{\tau}\right). \tag{41}$$

Focusing on the inequality:

$$\exp(-\alpha) \le \frac{1}{B-1} \exp\left(\frac{s_{ii}+1}{\tau}\right). \tag{42}$$

Multiply both sides by $B-1$:

$$(B-1)\exp(-\alpha) \le \exp\left(\frac{s_{ii}+1}{\tau}\right). \tag{43}$$

Take the natural logarithm of both sides:

$$\ln\left((B-1)\exp(-\alpha)\right) \le \frac{s_{ii}+1}{\tau}. \tag{44}$$

Simplify the left-hand side:

$$\ln(B-1) + \ln(\exp(-\alpha)) = \ln(B-1) - \alpha. \tag{45}$$

Thus:

$$\ln(B-1) - \alpha \le \frac{s_{ii}+1}{\tau}. \tag{46}$$

Multiply through by $\tau$:

$$\tau(\ln(B-1) - \alpha) \le s_{ii} + 1. \tag{47}$$

Finally, we have the lower bound for $s_{ii}$:

$$s_{ii} \ge \tau(\ln(B-1) - \alpha) - 1. \tag{48}$$

Thus, we have proved:

$$\mathrm{sim}\left(\varepsilon_k^i, \tilde{\varepsilon}_k^i\right) \ge \tau(\ln(B-1) - \alpha) - 1. \tag{49}$$

$$\square$$

## E   Proof of Theorem 1

**Theorem** (Continuity Upper Bound). *Suppose $\epsilon_\theta(\cdot)$ is L-Lipschitz with respect to z, and that $\tilde{z}_t^{k+1}$ evolves via the VJP of $\epsilon_\theta$. Then under the contrastive constraint $\mathcal{L}_{cont} \le \alpha$, we have:*

$$\|\tilde{z}_t^{k+1} - \tilde{z}_t^k\|_2^2 \le L^2 \|\epsilon_\theta(\hat{a}_t^k, z_t, k) - \epsilon_\theta(\hat{a}_t^k, \tilde{z}_t^k, k)\|_2^2 \le 2L^2(2 - \tau\ln(B-1) + \tau\alpha). \tag{50}$$

*Proof.* We split the proof into two steps:

**Step 1: Lipschitz continuity of $\epsilon_\theta$ relates latent updates and noise difference.** Taking the vector-Jacobian product (VJP) approximation for the drift of the SDE evolution, the update of $\tilde{z}_t^k$ at each step satisfies:

$$\|\tilde{z}_t^{k+1} - \tilde{z}_t^k\|_2 \ \propto \ \mathrm{VJP}(\hat{a}_t^k, z_t), \tag{51}$$

where, under the Lipschitz continuity assumption, the VJP depends linearly on the local noise gradient structure.

Assuming proper normalization of the step size in the SDE discretization (absorbed into the $L$-Lipschitz constant), we have:

$$\|\tilde{z}_t^{k+1} - \tilde{z}_t^k\|_2 \ \le \ L\|\epsilon_\theta(\hat{a}_t^k, z_t, k) - \epsilon_\theta(\hat{a}_t^k, \tilde{z}_t^k, k)\|_2. \tag{52}$$

Squaring both sides yields:

$$\|\tilde{z}_t^{k+1} - \tilde{z}_t^k\|_2^2 \ \le \ L^2 \|\epsilon_\theta(\hat{a}_t^k, z_t, k) - \epsilon_\theta(\hat{a}_t^k, \tilde{z}_t^k, k)\|_2^2. \tag{53}$$

**Step 2: Contrastive loss bounds the noise difference.** From Lemma 1, when $\mathcal{L}_{\mathrm{cont}} \le \alpha$, the cosine similarity between $\varepsilon_k^i$ and $\tilde{\varepsilon}_k^i$ is bounded below:

$$\mathrm{sim}\left(\varepsilon_k^i, \tilde{\varepsilon}_k^i\right) \ge \tau(\ln(B-1) - \alpha) - 1. \tag{54}$$

Recall that cosine similarity between two unit-norm vectors $u, v$ satisfies:

$$\|u - v\|_2^2 = 2(1 - \mathrm{sim}(u, v)). \tag{55}$$

Thus, applying this relation to $\varepsilon_k$ and $\tilde{\varepsilon}_k$, we obtain:

$$\|\epsilon_\theta(\hat{a}_t^k, z_t, k) - \epsilon_\theta(\hat{a}_t^k, \tilde{z}_t^k, k)\|_2^2 = 2\left(1 - \mathrm{sim}\left(\epsilon_\theta(\hat{a}_t^k, z_t, k), \epsilon_\theta(\hat{a}_t^k, \tilde{z}_t^k, k)\right)\right). \tag{56}$$

By plugging in the lower bound on similarity from Lemma 1, we get:

$$\|\epsilon_\theta(\hat{a}_t^k, z_t, k) - \epsilon_\theta(\hat{a}_t^k, \tilde{z}_t^k, k)\|_2^2 \leq 2(2 - \tau \ln(B - 1) + \tau\alpha). \tag{57}$$

Thus, combining Step 1 and Step 2, we can conclude that:

$$\|\tilde{z}_t^{k+1} - \tilde{z}_t^k\|_2^2 \ \leq \ L^2 \left\|\epsilon_\theta(\hat{a}_t^k, z_t, k) - \epsilon_\theta(\hat{a}_t^k, \tilde{z}_t^k, k)\right\|_2^2 \ \leq \ 2L^2(2 - \tau \ln(B - 1) + \tau\alpha). \tag{58}$$

This completes the proof. $\qquad\square$

# F  Robotic Manipulation Benchmarks

This ablation section details the tasks and corresponding datasets for four robotic manipulation benchmarks for simulation: Robomimic, Franka Kitchen, Push-T, and Dynamic Push-T. Each benchmark contributes unique tasks and datasets to evaluate various aspects of robotic manipulation, from precision and coordination to adaptability in dynamic environments. Below, we describe each task, its objectives, and the associated datasets, highlighting their roles in our empirical evaluations. **Robomimic Benchmark.** Robomimic is a comprehensive benchmark for robotic manipulation with five tasks: **Can**, **Square**, **Transport**, **Tool Hang**, and **Lift**. These tasks are designed to evaluate a range of manipulation skills that include pick-and-place, precision assembly, multi-arm coordination, and basic object manipulation. Each task includes expert demonstrations collected via proficient human (PH) teleoperation, with additional mixed human (MH) datasets, resulting in nine datasets total. The datasets are structured in HDF5 format, containing observations, actions, rewards, and other metadata, and are available online (`https://github.com/ARISE-Initiative/robomimic`).

Table 4: Summary of Robomimic Tasks and Datasets

| Task | PH Demos | MH Demos | Key Skills Tested |
|---|---|---|---|
| Can | 200 | 300 | Pick-and-place |
| Square | 200 | 300 | Precision manipulation |
| Transport | 200 (2 operators) | 300 | Multi-arm coordination |
| Tool Hang | 200 | 300 | Precision grasping, insertion |
| Lift | 200 | 300 | Basic object manipulation |

**Can.** The Can task involves picking up a can and placing it in the correct bin, testing pick-and-place skills. The dataset includes two variants:

- **Proficient Human (PH)**: 200 demonstrations collected by a single proficient operator using the RoboTurk platform (`https://roboturk.stanford.edu/`).

- **Mixed Human (MH)**: 300 demonstrations from six operators of varying proficiency, introducing variability in demonstration quality.

The PH dataset provides high-quality demonstrations, while MH introduces real-world variability.

**Square.** The Square task, also known as Square Nut Assembly, requires fitting a square nut onto a square peg (and potentially a round nut onto a round peg). The scene includes two colored pegs (square and round) and two nuts on a tabletop, with randomized nut locations at episode start (`https://robosuite.ai/docs/modules/environments.html`). This task evaluates precision manipulation. The dataset includes two variants:

- **Proficient Human (PH)**: 200 demonstrations collected by a proficient operator via RoboTurk.

- **Mixed Human (MH)**: 300 demonstrations from operators of varying proficiency.

These datasets test algorithms on precise manipulation with both expert and varied human inputs.

**Transport.** The Transport task involves transporting an object using two robotic arms, which requires multi-arm coordination. It is noted for its complexity, with observation spaces including shoulder and wrist views per arm, indicating a dual-arm setup (`https://robomimic.github.io/study/`). The dataset includes two variants:

- **Proficient Human (PH)**: 200 demonstrations collected by two proficient operators working together (`https://www.tensorflow.org/datasets/catalog/robomimic_ph`).
- **Mixed Human (MH)**: 300 demonstrations from varied operators.

The datasets evaluate coordination and robustness in multi-arm cooperation tasks.

**Tool Hang.** The Tool Hang task requires hanging a tool (e.g., a hook), which involves precise grasping and insertion. The dataset includes two variants:

- **Proficient Human (PH)**: 200 demonstrations collected by a proficient operator via RoboTurk.
- **Mixed Human (MH)**: 300 demonstrations from operators of varying proficiency.

These datasets evaluate on high-precision tasks.

**Lift.** The Lift task involves lifting an object, a simple manipulation task that benefits less from large datasets compared to complex tasks. The dataset includes two variants:

- **Proficient Human (PH)**: 200 demonstrations collected by a proficient operator via RoboTurk.
- **Mixed Human (MH)**: 300 demonstrations from operators of varying proficiency.

The datasets provide a baseline for evaluating basic manipulation skills.

Table 4 provides a summary of the Robomimic tasks and their corresponding datasets.

**Franka Kitchen Benchmark.** Franka Kitchen is a simulation benchmark where a 9-DoF Franka arm operates in a kitchen environment, performing four household tasks per trajectory. The tasks involve manipulating objects to achieve a desired goal configuration, which include:

- **Open the Microwave**: The robot opens the microwave door.
- **Move the Kettle**: The robot repositions the kettle to a target location.
- **Flip the Light Switch**: The robot toggles a light switch to turn on a light.
- **Slide Open the Cabinet Door**: The robot slides open a cabinet door.

The benchmark is hosted on platforms like Gymnasium-Robotics (`https://robotics.farama.org/envs/franka_kitchen/`) and uses datasets from D4RL (`https://github.com/Farama-Foundation/D4RL`).

Table 5: Summary of Franka Kitchen Tasks and Datasets

| Task | Description | Dataset | Key Skills Tested |
|------|-------------|---------|-------------------|
| Open Microwave | Open the microwave door | 566 trajectories | Object interaction, door manipulation |
| Move Kettle | Reposition the kettle | 566 trajectories | Object repositioning |
| Flip Light Switch | Toggle the light switch | 566 trajectories | Switch manipulation |
| Slide Cabinet Door | Slide open the cabinet door | 566 trajectories | Sliding mechanism interaction |

**Datasets.** The Franka Kitchen dataset consists of 566 trajectories, each completing all four tasks, collected via human teleoperation across seven interactive objects (a microwave, a kettle, an overhead light switch, a sliding cabinet, a hinged cabinet, a top burner, and a bottom burner) (`https://minari.farama.org/datasets/D4RL/kitchen/index.html`). The dataset evaluates the ability to sequence and execute multiple subtasks in a realistic kitchen environment. Table 5 provides a summary of the Franka Kitchen tasks and their corresponding datasets.

**Push-T.** Push-T is a manipulation task adapted from Implicit Broadcast Communication (IBC), where a circular end-effector pushes a T-shaped block to a target location. The block and end-effector start at random positions, which can be defined by different evaluation seeds. Observations are either RGB images (**img**) or nine 2D keypoints (**kp**) outlining the T block's shape and the target location, as well as the end-effector's position (`https://paperswithcode.com/task/robot-manipulation`).

**Dataset**: The Push-T dataset includes 62,500 push interactions across 200 evaluation seeds, designed to evaluate the pushing dynamics and perception in a controlled setting.

**Dynamic Push-T.** We extend the Push-T task by introducing a moving orange ball that travels at 100 units per second within a 500-unit square. The ball bounces off both the walls and the end-effector, occasionally disrupting the T-shaped block. The agent must simultaneously block these disturbances while pushing the block to the target, requiring it to maintain stable control under dynamic interference. This setting explicitly evaluates latent continuity and adaptability: abrupt latent shifts can destabilize interactions, whereas smooth latent evolution enables the agent to adapt effectively to changing dynamics.

**Dataset**: Similar to the Push-T dataset, we include 62,500 push interactions across 200 evaluation seeds, specifically designed to evaluate robustness and adaptability in dynamic and unpredictable environments. Table 6 provides a summary of the Push-T and Dynamic Push-T tasks and their corresponding datasets.

Table 6: Summary of Push-T and Dynamic Push-T Tasks and Datasets

| Task | Description | Dataset | Key Skills Tested |
|------|-------------|---------|-------------------|
| Push-T | Push T-shaped block to target | 62,500 pushes | Pushing dynamics and perception |
| Dynamic Push-T | Push T-shaped block with moving ball disturbance | 62,500 pushes | Adaptability and disturbance handling |

## G  Implementation Details

Our DP-AG builds on the Diffusion Policy (DP)[Chi et al., 2023], with two key modifications: (1) we add two linear heads for predicting the base drift $\mu_\phi(z_t)$ and log-variance $\log \sigma_\phi^2(z_t)$ from the static latent features, and (2) we incorporate VJP-guided SDE for latent evolution without introducing extra learnable parameters.

For observation encoding, we use a ResNet-18 [He et al., 2016] backbone for image-based tasks. We adopt the same conditional U-Net architecture from DP [Chi et al., 2023] to predict diffusion noise. During training, we apply random cropping with task-specific sizes as in DP, while a center crop is used at inference. To maintain dynamic consistency, no color jitter or random flipping is applied. An MLP is used to encode the agent's proprioceptive inputs. For key-point-based tasks, we follow the original DP setup and use fully connected networks.

We also apply action normalization scales each action dimension independently to $[-1, 1]$ to ensure compatibility with the DDPM denoising process, where predictions are clipped within this range at each step. For positional control tasks, actions use 6D rotation representations. Velocity control tasks use 3D axis-angle representations, which follow standard practice.

We train the model using the iDDPM algorithm [Nichol and Dhariwal, 2021] with 100 diffusion steps. All models are trained for 300 epochs on vision-based tasks and 200 epochs for key-point-based tasks. For learning rate scheduling, we use a cosine annealing schedule with a linear warmup of 500 steps. Batch sizes are set to 64 for image-based tasks and 256 for key-point-based tasks. Optimization uses AdamW with a learning rate of $1 \times 10^{-4}$ in all experiments. All hyperparameters not directly related to DP-AG extensions (e.g., diffusion step count, augmentation, action normalization) are kept identical to DP for a controlled and fair comparison. For inference, we maintain the same number of diffusion denoising steps as training to avoid introducing a distributional shift.

## H  Ablation Studies

### H.1  Latency and Computational Cost Analysis

We evaluate the computational overhead introduced by our DP-AG compared to the original DP by measuring both training time per epoch, the inference latency, and the total training time to convergence. The computation analysis is conducted on the Push-T benchmark. Experiments are conducted on an Nvidia RTX 4090 GPU with 24GB VRAM. Table 7 summarizes the detailed computational cost comparison between the DP baseline and our DP-AG.

**Training Time per Epoch.** On the Push-T benchmark, the average training time per epoch increases slightly from 114.2 seconds for DP to 119.5 seconds for DP-AG, corresponding to an overhead of 4.6%. This increase is expected because VJP computation requires additional backward-mode automatic differentiation through the noise predictor. However, the extra cost remains moderate, largely due to the relatively small dimensionality of the latent observation space.

**Inference Latency.** At inference time, the VJP computations are omitted entirely. Our DP-AG simply operates with the latent features extracted from the observation encoder at inference. As a result, inference latency remains virtually unchanged: 145.3 milliseconds per action sequence generation for DP versus 146.5 milliseconds for our DP-AG, a marginal 0.8% difference. This negligible overhead ensures that DP-AG maintains real-time responsiveness for high-frequency robotic control while improving the smoothness and consistency of action generation.

**Training Efficiency and Total Time to Convergence.** Although our DP-AG introduces a minor increase in per-epoch training time, it significantly accelerates convergence. On the Push-T benchmark, DP requires approximately 200 epochs to converge, whereas DP-AG achieves comparable performance within only around 100 epochs. This effectively reduces the number of required training epochs and the total training time by nearly 50%. To be specific, DP completes training in about 6.2 hours, while DP-AG completes training in approximately 3.5 hours, which results in a net saving of 2.7 hours. Thus, despite the slight per-epoch overhead, our DP-AG achieves faster overall training and improved sample efficiency.

**Summary.** Therefore, while DP-AG introduces additional VJP computations, modern autodiff frameworks (*e.g.*, PyTorch) execute them efficiently, leading to negligible runtime overhead. As shown in Table 7, DP-AG incurs only a minor 4.6% increase in per-epoch training time but delivers nearly 50% higher training efficiency. Inference latency remains virtually unchanged: DP-AG sustains real-time control on the UR5 robot while producing smoother trajectories, lower jerk, and faster task completion. These results demonstrate that the added computation does not hinder deployment or responsiveness, and that our perception–action interplay both improves DP and substantially accelerates training, making DP-AG well suited for real-world applications where rapid retraining and real-time control are important.

Table 7: Computational cost comparison between DP and DP-AG on Nvidia RTX 4090 (Push-T).

| Model | One Epoch Time (s) | Epochs to Converge | Total Training Time (h) | Inference Latency (ms) |
|---|---|---|---|---|
| DP | 114.2 | $\sim$200 | $\sim$6.2 | 145.3 |
| DP-AG (Ours) | 119.5 | $\sim$100 | $\sim$3.5 | 146.5 |

### H.2  Effect of Cycle-Consistent Contrastive Loss

To evaluate the importance of the cycle-consistent contrastive loss in our DP-AG, we conduct an ablation study by removing this component while keeping the rest of the architecture unchanged. This allows us to isolate the role of cycle consistency in enforcing mutual smoothness between latent evolution and action refinement during training.

**Setup.** We compare two variants on the Push-T benchmark:

- DP-AG (full model): Includes the cycle-consistent contrastive loss between static and VJP-guided noise predictions.
- DP-AG w/o Contrastive: Removes the contrastive term from the training objective, relying only on the diffusion loss and KL regularization.

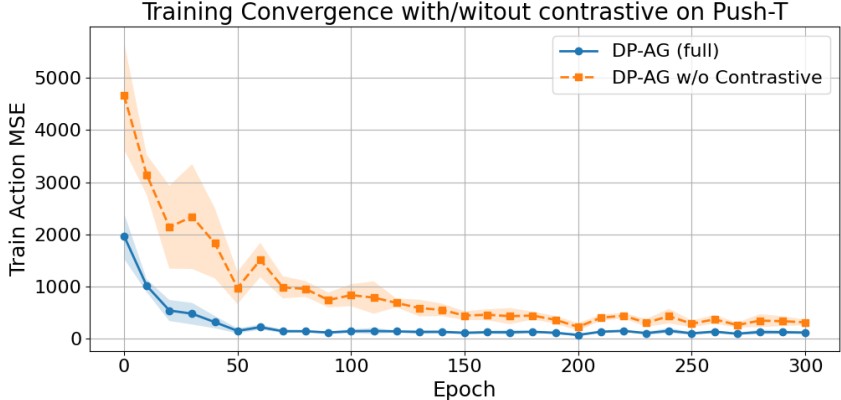

Figure 7: Comparison of training convergence on the Push-T benchmark for DP-AG with and without cycle-consistent contrastive loss.

Both variants are trained under the same settings; and the action MSE is evaluated across training epochs. We repeat the experiments with 5 different initialization seeds for training.

**Results.** As shown in Figure 7 and Table 8, removing the cycle-consistent loss leads to slower convergence, higher final training action MSE, and reduced target coverage scores. To be specific, the model without contrastive loss requires approximately 3 times more epochs to converge compared to the full DP-AG; the final success rate drops from 93% to 85% on the Push-T benchmark; and the training action MSE remains consistently higher throughout training, which indicates less accurate action generation.

**Analysis.** Without the contrastive alignment between evolving and static latents, VJP-guided perturbations can drift away from the optimal action refinement trajectory, which degrades both latent continuity and action smoothness. The cycle-consistent loss is important in closing the perception-action loop, which can ensure that latent evolution remains tightly coupled with action denoising across diffusion steps.

Table 8: Effect of cycle-consistent contrastive loss on Push-T benchmark.

| Model | Epochs to Converge | Converged Train Action MSE | Score |
|---|---|---|---|
| DP-AG (full) | 100 | 65.8 | 0.93 |
| DP-AG w/o Contrastive | 300 | 183.5 | 0.85 |

### H.3 Likelihood Supervision vs. Contrastive Loss

In Section 5.1, we derive a noise regression objective from the variational lower bound, which serves a similar role to our cycle-consistent contrastive loss. To evaluate the necessity of the contrastive loss, we replace it with the likelihood-based noise regression objective defined in Equation 13:

$$\mathcal{L}_{\text{LH}} = \mathbb{E}_{(o_t, a_t) \sim \mathcal{D}, \, k \sim \mathcal{U}(1, K)} \left[ \| \epsilon_\theta(\hat{a}_t^k, \tilde{z}_t^k, k) - \epsilon \|_2^2 \right]. \tag{59}$$

**Experimental Setup.** We compare three variants:

- **Cycle-Consisteny Contrastive**: Uses the proposed cycle-consistent contrastive loss $\mathcal{L}_{\text{cont}}$ only.
- **MSE-Only**: Replaces $\mathcal{L}_{\text{cont}}$ with the noise regression loss $\mathcal{L}_{\text{LH}}$.
- **Combined**: Combines both objectives: $\mathcal{L} = \mathcal{L}_{\text{cont}} + \mathcal{L}_{\text{LH}}$.

All models are trained with 5 random seeds on the Push-T benchmark. We report the mean and standard deviation of validation accuracy over 100 epochs.

**Results.** As shown in Figure 8, the contrastive-only variant not only converges faster but also achieves the best validation accuracy. Although MSE yields smoother convergence, it lacks the structural

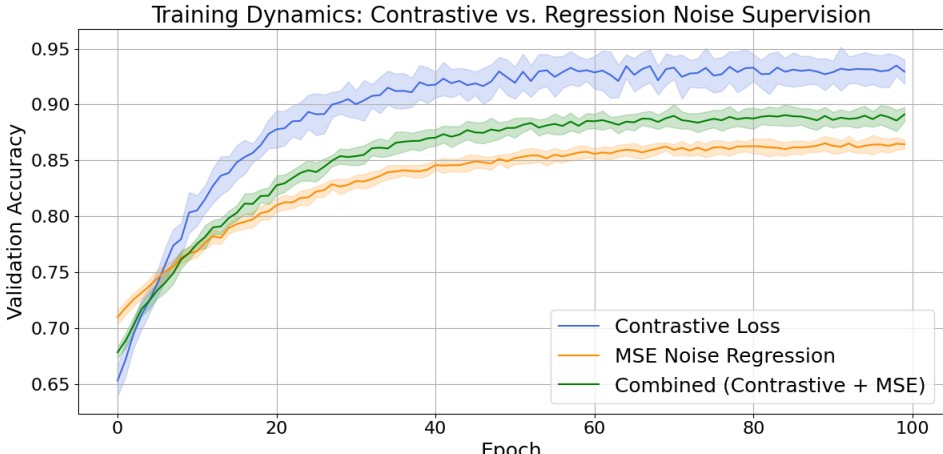

Figure 8: Validation accuracy during training with contrastive loss vs. MSE noise regression on the Push-T benchmark. Contrastive loss achieves faster convergence and better performance but with slightly more variance during training.

benefits of noise alignment that contrastive learning offers. The combined objective does not improve performance over the contrastive-only variant and sometimes results in unstable training, likely due to conflicting optimization signals between absolute and relative supervision. These results validate that the contrastive alignment provides stronger inductive bias for learning dynamic consistency between static and VJP-guided latents, while the likelihood term, though informative, introduces redundancy. We thus omit the likelihood objective from our final model.

### H.4 Effect of VJP Strength

In this section, we study how the strength of the VJP-guided perturbations affects the performance of our DP-AG. Recall that the VJP acts as a stochastic "force" that shapes the evolution of latent observation features based on the diffusion process that refines the action generation. While moderate VJP guidance helps structure latent trajectories coherently, overly weak or strong guidance may destabilize training or restrict flexibility.

**Setup.** We introduce a scaling factor $\gamma$ applied to the VJP term during latent updates in Equation 8:

$$dz_t = \gamma \cdot \text{VJP}(\hat{a}_t^k, z_t)\, dt + \sigma_\phi(z_t)\, dW_t, \tag{60}$$

We evaluate the following settings: $\gamma \in \{0.0,\ 0.5,\ 1.0,\ 2.0,\ 5.0\}$, where $\gamma = 1.0$ is our default setting. We repeat the experiments with 5 different initialization seeds for training.

**Results.** The results are presented in Figure 9 and Table 9. We observe that:

- Without VJP guidance, latent features remain static during diffusion, effectively reducing our DP-AG to a standard DP.

- Moderate VJP strength ($\gamma = 0.5, 1.0$) achieves the best results, which tradeoff a balance between encouraging latent evolution and maintaining stability.

- High VJP strength ($\gamma = 2.0, 5.0$) leads to unstable latent evolution, where the updated trajectories deviate excessively from the static latent features, ultimately degrading policy performance.

### H.5 Effect of KL Loss Hyperparameter

In this section, we investigate the effect of varying the KL divergence coefficient $\lambda_{\text{KL}}$ on the performance of our DP-AG using the Push-T benchmark. While $\beta$-VAE [Higgins et al., 2017] tunes KL regularization to promote disentanglement, in our case, the KL term stabilizes the action-guided latent evolution without affecting its ability to adaptively track action refinements.

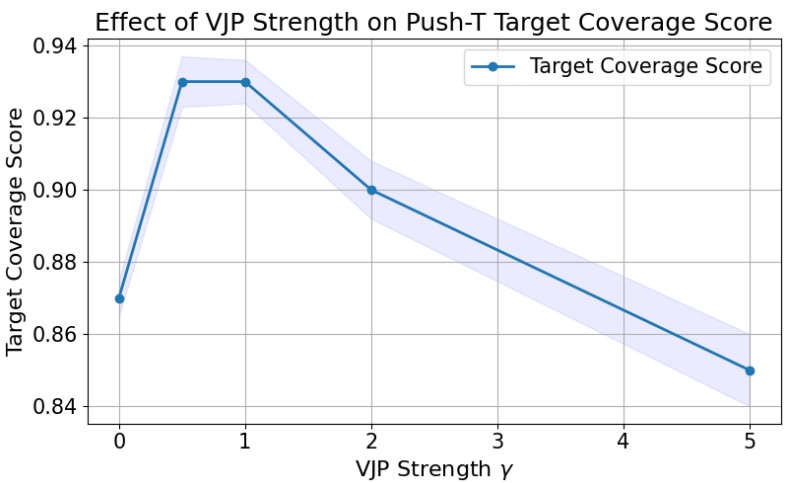

Figure 9: Effect of the VJP strength $\gamma$ on the Push-T benchmark.

Table 9: Effect of VJP strength $\gamma$ on Push-T benchmark.

| $\gamma$ | Final Train Action MSE | Score | Latent Behavior |
|---|---|---|---|
| 0.0 | 75.3 | 0.87 | static |
| 0.5 | 58.3 | 0.93 | smooth |
| 1.0 | 65.8 | 0.93 | smooth |
| 2.0 | 71.2 | 0.90 | over-reactive |
| 5.0 | 241.5 | 0.85 | unstable |

**Setup.** We train DP-AG on the Push-T benchmark with the following $\lambda_{\text{KL}}$ values: $\{0, 0.5, 1.0, 2.0, 4.0, 5.0, 10.0\}$. All other training configurations remain unchanged across experiments. We repeat the experiments with 5 different initialization seeds for training.

**Results.** The results are presented in Figure 10 and summarized in Table 10. We observe the following trends:

- Without KL regularization ($\lambda_{\text{KL}} = 0$), latent evolution becomes less stable, causing updates to drift excessively from the static latent and resulting in lower target coverage scores.

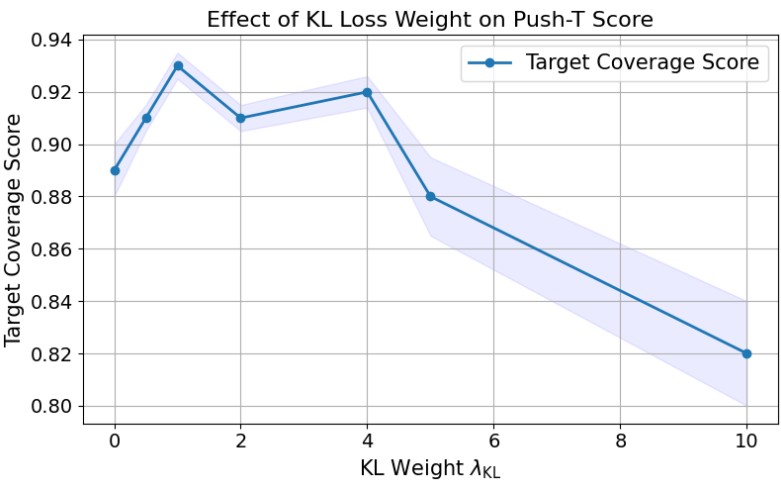

Figure 10: Effect of KL loss weighting $\lambda_{\text{KL}}$ on the target coverage score for the Push-T benchmark.

Table 10: Effect of KL loss hyperparameter $\lambda_{\text{KL}}$ on Push-T benchmark.

| $\lambda_{\text{KL}}$ | Converged Train Action MSE | Score | Latent Stability |
|---|---|---|---|
| 0 | 123.1 | 0.89 | slightly unstable |
| 0.5 | 86.6 | 0.91 | stable |
| 1.0 | 65.8 | 0.93 | stable |
| 2.0 | 91.3 | 0.91 | slightly over-constrained |
| 4.0 | 78.3 | 0.92 | over-constrained |
| 5.0 | 168.8 | 0.88 | over-constrained |
| 10.0 | 324.5 | 0.82 | severely over-constrained |

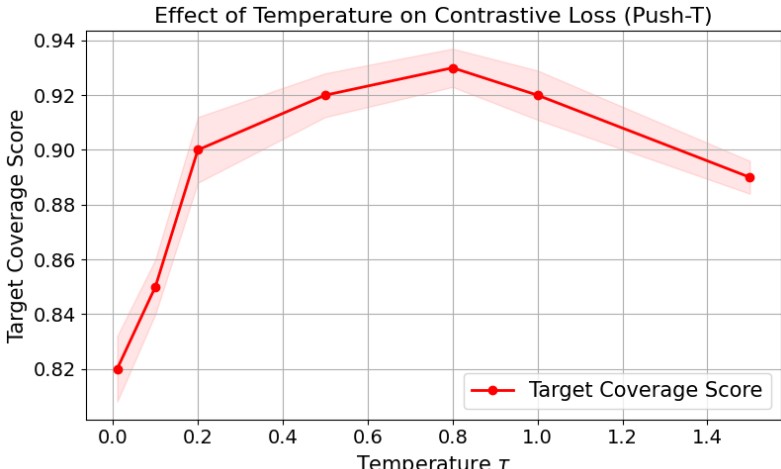

Figure 11: Effect of temperature $\tau$ in the cycle-consistent contrastive loss on Push-T benchmark performance.

- Small KL values ($\lambda_{\text{KL}} = 0.5, 1.0$) achieve the best trade-off between flexibility and stability, which leads to the best performance.

- Moderate to large KL values ($\lambda_{\text{KL}} = 2.0, 4.0, 5.0$) begin to over-regularize the latent space, which limits its ability to adapt to action refinements.

- Strong KL regularization ($\lambda_{\text{KL}} = 10.0$) severely restricts latent evolution, which causes underfitting and significantly lower target coverage score.

**Analysis.** These results suggest that the KL regularization in our DP-AG should not be viewed through the lens of promoting disentanglement, as in $\beta$-VAE. Instead, it acts to anchor the VJP-guided latent dynamics toward stable yet adaptive evolution.

## H.6 Effect of Contrastive Loss Temperature Parameter

In this section, we investigate the influence of the temperature parameter $\tau$ in the cycle-consistent contrastive loss on the performance of our DP-AG. As in self-supervised learning settings [Chen et al., 2020], the temperature parameter controls the sharpness of similarity scores between a query and its positive pairs in the contrastive learning. A low $\tau$ enforces highly sharp alignment, while a high $\tau$ leads to smoother matching across samples.

In our DP-AG, unlike its usage for self-supervised learning, our contrastive loss is designed to enforce cycle consistency between perception and action noise predictions during latent evolution. To evaluate the impact of the temperature parameter on this novel perception-action interplay, we vary $\tau$ across a broad range: $\tau \in \{0.01, 0.1, 0.2, 0.5, 0.8, 1.0, 1.5\}$. We repeat the experiments with 5 different initialization seeds for training.

**Results.** Figure 11 presents the results, with a summary provided in Table 11. We observe:

- Very low temperature ($\tau = 0.01,\ 0.1$) could overly sharp alignment that causes unstable training and brittle latent evolution.

- Moderate temperatures ($\tau = 0.2,\ 0.5$) improve stability compared to $\tau = 0.01$, but still do not fully leverage latent adaptability.

- High temperature ($\tau = 0.8$) achieves the best performance, which yields the highest success rates and smoothest latent evolution.

- Very high temperature ($\tau = 1.5$) could weaken cycle consistency, which leads to performance degradation.

Table 11: Effect of temperature $\tau$ on cycle-consistent contrastive loss (Push-T benchmark).

| $\tau$ | Converged Train Action MSE | Score | Stability |
|---|---|---|---|
| 0.01 | 433.8 | 0.82 | brittle |
| 0.1 | 243.4 | 0.85 | unstable |
| 0.2 | 186.9 | 0.90 | improved |
| 0.5 | 103.1 | 0.92 | stable |
| 0.8 | 65.8 | 0.93 | best stability |
| 1.0 | 96.3 | 0.92 | slightly diffuse |
| 1.5 | 105.5 | 0.89 | diffuse |

**Analysis.** In our DP-AG, a higher temperature (e.g., $\tau = 0.8$) provides better flexibility for aligning noise predictions during latent evolution, allowing dynamic perception refinement while maintaining cycle consistency. Extremely low $\tau$ values over-constrain the model, while excessively high values ($\tau > 1.0$) can weaken the noise prediction alignment.

### H.7 Comparison with Input Perturbation Smoothness

To evaluate whether the stability achieved by DP-AG could be reproduced by alternative smoothness regularization, we compared against baselines that enforce consistency under input perturbations. For each observation, we generated a perturbed version by adding Gaussian noise, then minimized MSE or cosine similarity between their predicted action noise scores. Table 12 reports results on the Push-T benchmark. Action smoothness is quantified as normalized inverse jerk (higher is smoother).

**Results.** DP-AG clearly outperforms perturbation-based baselines, achieving higher success rates and smoother action sequences. This demonstrates that contrastive regularization not only enforces smoothness but also preserves semantic alignment, leading to superior policy performance.

Table 12: Comparison of DP-AG with input perturbation smoothness baselines on Push-T.

| Method | SR (img) | SR (kp) | Smoothness (img ↑) | Smoothness (kp ↑) |
|---|---|---|---|---|
| Perturbation MSE | 0.85 | 0.92 | 0.83 | 0.87 |
| Perturbation Cosine | 0.88 | 0.95 | 0.86 | 0.92 |
| **DP-AG (Contrastive Loss)** | **0.93** | **0.99** | **0.91** | **0.95** |

## I More Comparisons of Training Convergence

In this section, we compare the training convergence behavior between the baseline DP and our DP-AG in Robomimic Lift and Square tasks. Training convergence is evaluated by measuring the MSE between predicted and ground-truth actions over training epochs.

**Setup.** Both the baseline DP and our DP-AG models are trained under the same settings, which use the same dataset splits, optimizer configurations, and diffusion schedules, etc. Training is conducted on an Nvidia RTX 4090 GPU with a batch size of 64, and convergence is measured by tracking the action MSE over 300 epochs for each method.

**Results.** The convergence curve for the Robomimic Square task is shown in Figure 12, and the curve for the Robomimic Lift task is shown in Figure 13. These results demonstrate that the action-guided

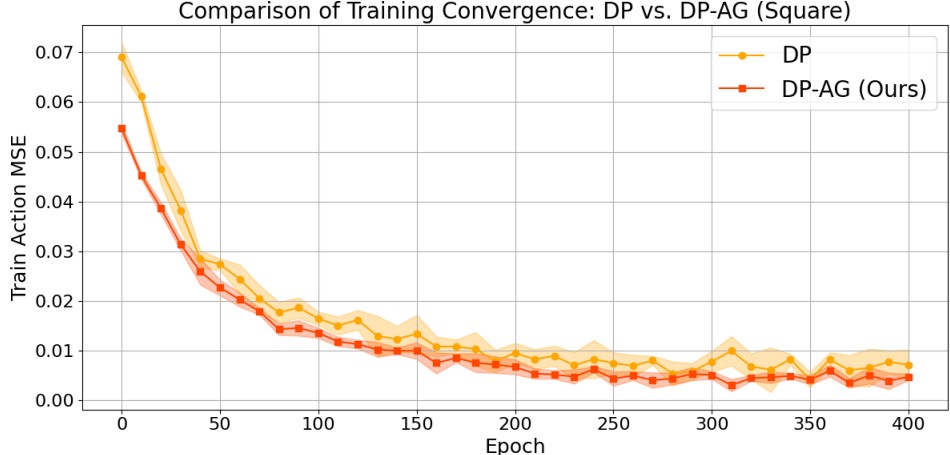

Figure 12: Training action MSE over epochs on Robomimic Square.

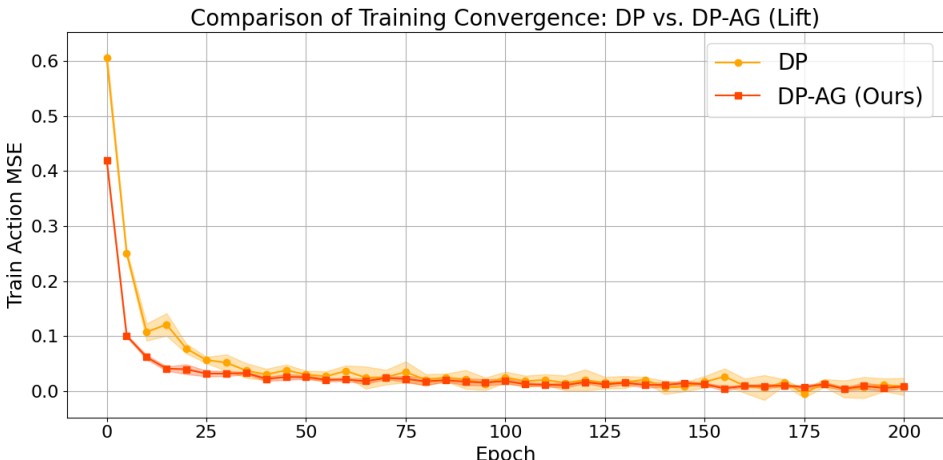

Figure 13: Training action MSE over epochs on Robomimic Lift.

perception updates in our DP-AG not only improve the policy performance but also enhance the training dynamics. By dynamically refining latent features through VJP-guided evolution, DP-AG accelerates the denoising process, which leads to more sample-efficient and stable training compared to the static latent used in the baseline DP.

## J  Real-World Manipulation Experiments on UR5 Robotic Arm

We conduct real-world manipulation experiments using the Universal UR5 robotic arm (`https://www.universal-robots.com/products/ur5e/`), a 6-degree-of-freedom platform widely adopted in both industrial and research contexts for its versatility and reliability. In this section, we compare the performance of our proposed DP-AG against the baseline Diffusion Policy (DP) across three visuomotor tasks: Painting, Candy Push, and Peg-in-Hole Insertion. These tasks are specifically designed to evaluate the perception–action interplay, testing each policy's ability to generalize from limited sensory inputs (e.g., RGB images) to precise motor control commands. All evaluations are conducted in a controlled real-world setting, allowing us to evaluate each method's effectiveness, robustness, and potential for the real-world deployment in practical robotic applications.

**Dataset Collection.** We prepared task-specific datasets for Painting, Candy Push, and Peg-in-Hole Insertion to support our real-world experiments using the UR5 robotic arm. Each dataset was collected

via kinesthetic teaching using a teach pendant. Human operators manually guided the robot through the desired trajectories, while synchronized visual and proprioceptive data were recorded in real time.

RGB images were captured at 30 frames per second (FPS) using two calibrated cameras: one fixed in front of the workspace and another mounted on the robot wrist to provide egocentric views. Simultaneously, robot joint states and Cartesian end-effector poses were recorded at 100 Hz via the UR5's internal API. To ensure alignment between visual and action modalities, timestamps from both the camera streams and robot sensors were synchronized using a shared system clock. Each trajectory thus consisted of time-aligned sequences of images and robot poses.

For the Painting task, we collected 100 trajectories by having the operator trace either heart-shaped or circular paths on a horizontal plane. Ground-truth 2D target contours were predefined, and the demonstrations were guided to closely follow these shapes. The Candy Push task involved 80 trials where small candy objects were randomly placed within a $20\text{cm} \times 20\text{cm}$ area. The human operator manually guided the robot to push each candy toward a designated target area marked on the table surface. Success or failure labels were assigned based on whether all candies reached the target area within a tolerance threshold. In the Peg-in-Hole Insertion task, 50 demonstrations were gathered by inserting pegs into variable hole positions, with occlusion and visual ambiguity intentionally introduced. Success was defined as full insertion without slippage or bounce-back. All demonstrations were segmented and temporally synchronized with the image streams using trajectory timestamps. Each dataset was then split into 80% of the trajectories for training and 20% for evaluation.

**Training and Implmentation Details.** The training procedure and implementation closely follow those used in the simulation benchmark experiments. We build our training pipeline on top of the DP framework, incorporating two core changes in our DP-AG: (1) two separate linear projection heads are added to map the static latent features $z_t$ to the drift term $\mu_\phi(z_t)$ and log-variance $\log \sigma_\phi^2(z_t)$, and (2) latent evolution is guided using a Vector–Jacobian Product (VJP)-driven stochastic differential equation (SDE), implemented without introducing new trainable parameters.

We extract visual features from RGB frames using a pre-trained ResNet-152 encoder. Proprioceptive inputs, including joint angles and velocities, are processed through a two-layer MLP. For diffusion noise prediction, we use the same conditional U-Net architecture as in the simulation experiments, where white noise serves as input and observation features act as conditioning signals. During training, we apply random cropping with sizes tuned per task; at evaluation time, we apply center cropping for consistency. No color jitter or random horizontal flipping is used, as such augmentations may disrupt the temporal consistency critical for dynamics modeling.

Actions are normalized to the $[-1, 1]$ range per dimension to align with the denoising diffusion process, and outputs are clipped to stay within bounds. Peg-in-Hole tasks use 6D continuous rotation representations to avoid gimbal lock, while Candy Push and Painting use 3D axis-angle vectors for velocity control. Training follows the iDDPM objective with 100 diffusion steps. Each policy is trained for 200 epochs with a batch size of 64, using AdamW optimization and a cosine learning rate schedule with a 500-step warmup and a base learning rate of $1 \times 10^{-4}$.

**Evaluation Metrics.** We evaluated each policy using 20% held-out trajectories per task, with each trial repeated 20 times. Four task-specific metrics were used to evaluate performance: success rate, smoothness (average jerk), IoU (for Painting), and time to complete.

- **Success Rate (%)**: Measures how often the robot completes the task correctly among all trials. In Candy Push, a trial is successful if all candies are pushed into the target zone. For Peg-in-Hole, success means the peg is fully inserted. We report mean and standard deviation over 20 trials to demonstrate consistency under variations like random object positions or occlusions.

- **Smoothness (Average Jerk)**: Captures the quality of motion based on jerk (the rate of change of acceleration). It is derived from the end-effector's position data (recorded at 100 Hz) using finite difference approximations. For a position sequence $\{x_1, x_2, \ldots, x_n\}$ at uniform time intervals $\Delta t$, average jerk is computed as:

  Velocity (first derivative):

$$v_i = \frac{x_{i+1} - x_i}{\Delta t}. \tag{61}$$

  Acceleration (second derivative):

$$a_i = \frac{v_{i+1} - v_i}{\Delta t} = \frac{x_{i+2} - 2x_{i+1} + x_i}{\Delta t^2}. \tag{62}$$

Jerk (third derivative):

$$j_i = \frac{a_{i+1} - a_i}{\Delta t} = \frac{x_{i+3} - 3x_{i+2} + 3x_{i+1} - x_i}{\Delta t^3}. \tag{63}$$

Average jerk across the trajectory:

$$\text{Average Jerk} = \frac{1}{n} \sum_{i=1}^{n} \|j_i\|. \tag{64}$$

Lower jerk values indicate smoother movements, which are especially important in tasks like Painting and insertion.

- **Intersection over Union (IoU in %)**: Used only in Painting, IoU evaluates how well the painted shape matches the ground-truth contour (i.e., heart, circle), which is calculated as:

$$\text{IoU} = \frac{\text{Intersection Area}}{\text{Union Area}}. \tag{65}$$

A higher IoU means better adherence to the intended shape.

- **Time to Complete (s)**: Records how long it takes to finish the task, which evaluates the efficiency alongside accuracy and smoothness. Faster times, particularly in Candy Push, suggest better adaptation to object dynamics.

These tasks evaluate perception-action interplay and generalization from partially observable inputs. Painting demands smooth motion, where jerk and IoU are important. Candy Push tests adaptation to varying layouts, with success rate and time as the key indicators. Peg-in-Hole requires 3D reasoning from 2D data, where success rate and jerk highlight stability. The dataset split ensures robust training and evaluation, consistent with the standard practice. In future work, we plan to explore sim-to-real transfer for our DP-AG, enabling real-world deployment with minimal or no additional data collection or model retraining.

## K  Visualizations of Action-Guided Latent Evolution

To clarify the semantic meaning of the latent drift in our DP-AG and why it matters for action prediction, we visualize how observation latents evolve under action guidance on the Push-T benchmark. The key question is: *"If the end-effector must move toward its desired future state, how should the latent representation of the current scene be adjusted?"* In DP-AG, the answer is given by the VJP, which nudges the latent toward features that are most predictive of the next action. This latent drift is not arbitrary. It is shaped by the policy's uncertainty: when the action head is confident, updates are small; when the action is ambiguous, the latent is pulled toward features that disambiguate the correct motion. Thus, drift is expected to improve prediction by refining perception precisely where it is action-relevant.

To make these refinements interpretable, we decode the evolving latents with a lightweight VAE. This choice directly addresses the concern of whether latent drift has semantic meaning: decoded frames expose what the policy is implicitly "re-seeing" as actions unfold. If latent evolution were meaningless noise, decoded frames would be incoherent; instead, we observe structured changes that align with the intended push trajectory.

### K.1  Decoded Reconstructions and Policy Attention.

**Implementation.** Let $z_0$ be the static latent from the encoder. At each VJP step $k$, we update $z_k$ with an action-aligned perturbation normalized to fixed step size. A lightweight VAE, trained only on static latents and then frozen, decodes each $z_k$ to an image $\tilde{x}_k$. Gradients are never propagated back through this decoder. To reveal the policy's evidence, we also compute Grad-CAM on the $(x, y)$ end-effector head and overlay it on the decoded frames.

**Visualization.** Figure 14 shows that the decoded VJP-evolved latents yield localized and coherent changes: the end-effector (blue gripper) shifts step by step toward its actual future position, while the background remains stable. Grad-CAM overlays (Figure 15) confirm that these same regions receive the strongest policy attention. Notably, the static latent ($k = 0$) contains no future cue,

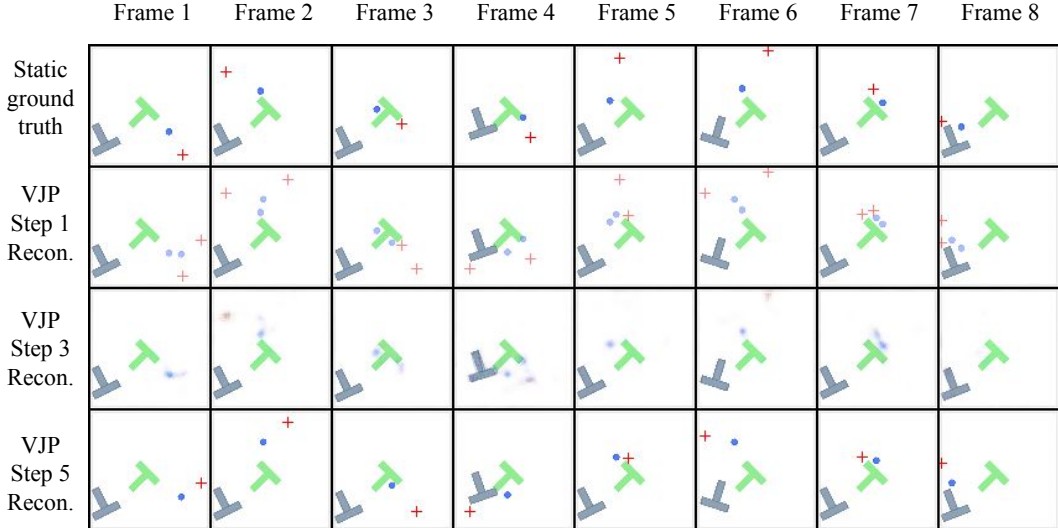

Figure 14: **Decoded VJP-evolved observation latents.** Columns are independent episodes. The top row ($k{=}0$) decodes the static latent. Rows $k{=}1,\ldots,5$ decode the latent after each VJP step.

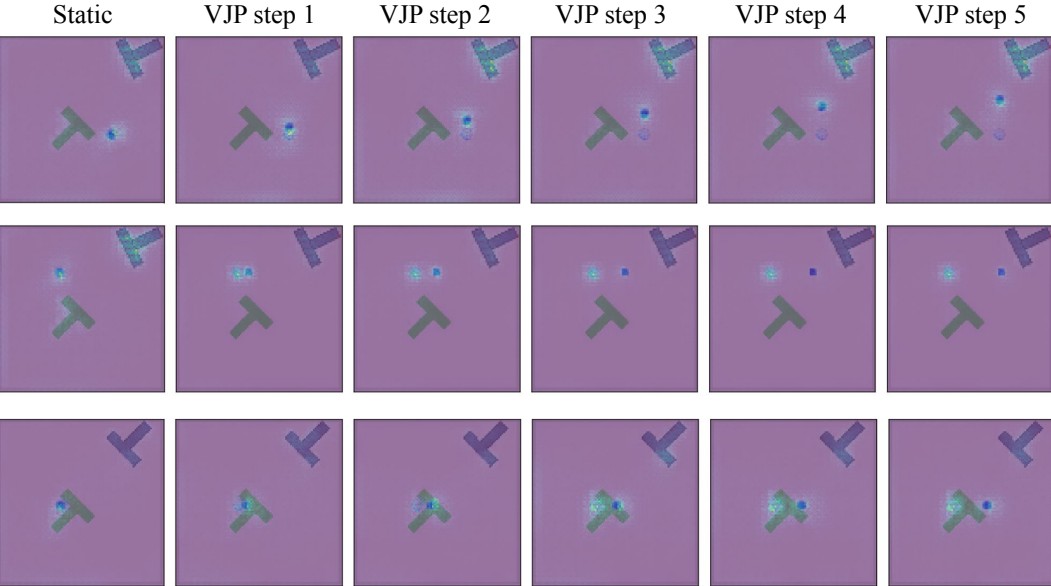

Figure 15: **Policy evidence on decoded frames.** Grad-CAM from the end-effector $(x, y)$ head of the policy is overlaid on the same reconstructions as Figure 14.

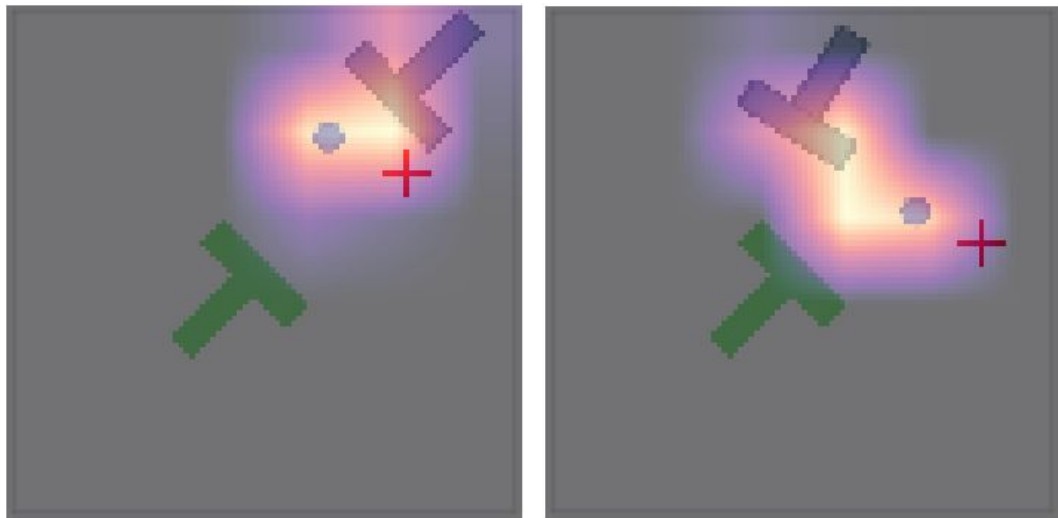

Figure 16: **Grad-CAM (control view).** Spatial evidence of the policy without reconstructions.

whereas the evolved latents ($k \geq 1$) highlight the precise spatial displacements the policy will execute. This demonstrates that the action-guided latent drift has both semantic meaning (visible in decoded images) and functional value (aligned with the policy's attention).

Together, the two figures demonstrate that VJP-induced latent drift produces interpretable image-space refinements and that these refinements are spatially aligned with the policy's own predictions. This coupling connects perceptual updates directly to the end-effector objective, demonstrating how action-guided latent evolution sharpens the visual grounding of control.

### K.2 Attention-Only Policy View.

As a control, we project Grad-CAM directly onto the raw input frames without decoding (Figure 16). These heatmaps still trace the end-effector's trajectory, confirming that the decoder does not hallucinate the effect. Instead, the VAE reconstructions provide a semantic lens: they make explicit how the evolving latent *imagines* the scene differently after each update.

### K.3 t-SNE of Action-Guided Latent Evolution.

While decoded frames and Grad-CAM demonstrate where VJP modifies the observation in pixel space, they do not reveal how these updates are organized in latent space. t-SNE embeddings provide a complementary view by projecting both static and VJP-evolved latents into two dimensions, allowing us to inspect whether action-guided updates follow structured and task-aligned trajectories rather than random drift.

**Visualization.** The t-SNE embeddings in Figure 17 show that VJP does not cause latents to scatter randomly. Instead, each static latent ($k{=}0$, blue) serves as an anchor from which the evolved latents ($k{=}1,\ldots,5$) trace short, smooth, and consistently oriented trajectories. Across episodes, these trajectories align to form clusters that correspond to motion modes, such as push direction, indicating that VJP structures the latent space around action-relevant dynamics rather than noise. Importantly, the cycle-consistent contrastive loss further reinforces this structure: it pulls each evolved latent back toward its static anchor while pushing it away from unrelated samples, preserving semantic grounding and preventing arbitrary drift. As a result, the trajectories remain compact and discriminative, ensuring that latent evolution not only tracks action dynamics but also maintains clear and task-relevant separation.

In summary, across decoded frames, attention overlays, and t-SNE latent embeddings, a coherent picture emerges:

1. VJP-driven drift produces semantically interpretable refinements (decoded end-effector positional shifts).

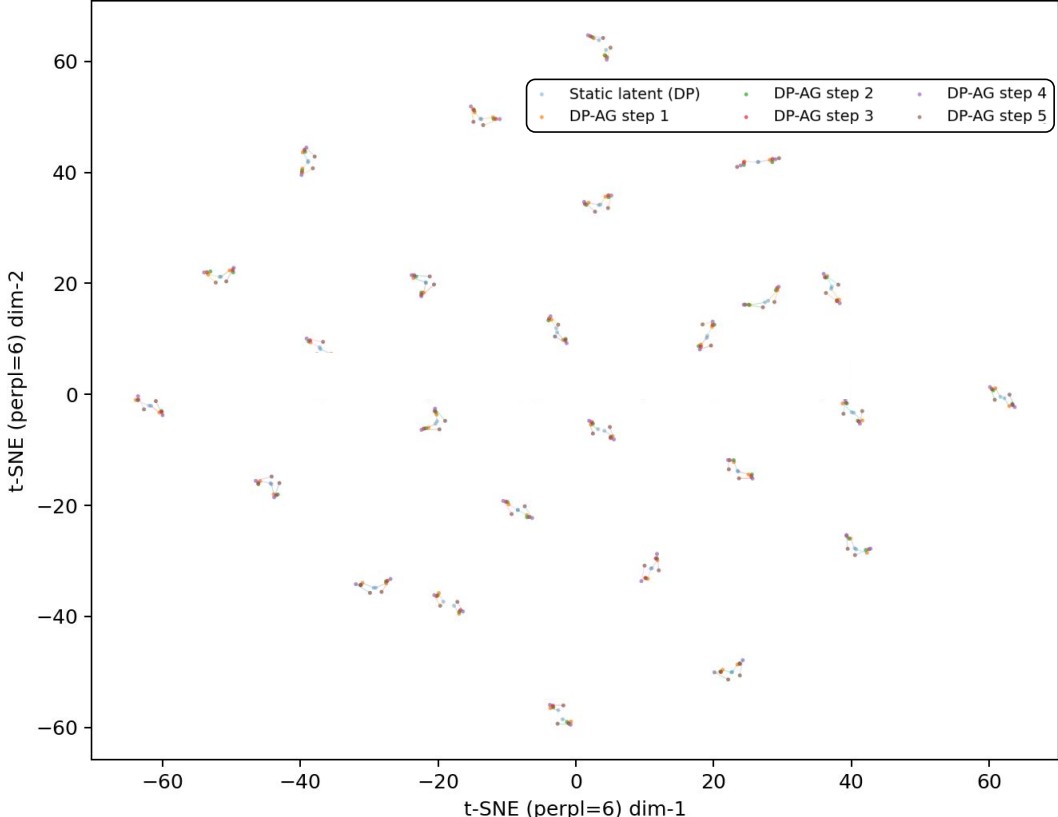

Figure 17: **t-SNE of static vs. evolved latents.** Each static latent (blue) is paired with its VJP trajectory over five steps (orange → green → red → purple → brown).

2. These refinements highlight exactly the regions that the policy deems most informative for predicting the next action.

3. In latent space, drift follows smooth and task-aligned trajectories rather than arbitrary noise.

Together, these results demonstrate that action-guided latent evolution is not only mathematically principled but also semantically and functionally meaningful.

## L   Extension to Online Mode

Although our main experiments are conducted in offline imitation learning, DP-AG is inherently well-suited for online training. In this section, we present a streaming variant of DP-AG that remains fully consistent with the original formulation. Its key mechanism, action-guided latent evolution via VJP, naturally supports the incremental refinement of perceptual features as new action-observation pairs arrive. Below, we describe the specific modifications that enable DP-AG to function effectively in the online regime:

**Online latent evolution.** At each diffusion step $k \in \{1, \ldots, K\}$ between $t$ and $t+1$, the latent observation is updated using the same VJP-guided rule as in the main model:

$$\tilde{z}_t^k = \mu_\phi(z_t) + \gamma \, \sigma_\phi(z_t) \odot \left( \frac{\partial \epsilon_\theta(\hat{a}_t^k, z_t, k)}{\partial z_t} \right)^\top \epsilon_\theta(\hat{a}_t^k, z_t, k), \tag{66}$$

where $\gamma$ controls the step size of action-guided latent refinement. Noise predictions conditioned on static and evolved latents are

$$\varepsilon_k = \epsilon_\theta(\hat{a}_t^k, z_t, k), \qquad \tilde{\varepsilon}_k = \epsilon_\theta(\hat{a}_t^k, \tilde{z}_t^k, k),$$

and are aligned through the cycle-consistent InfoNCE loss

$$\mathcal{L}_{\text{cont}} = -\frac{1}{B} \sum_{i=1}^{B} \log \frac{\exp\big(\text{sim}(\varepsilon_k^i, \tilde{\varepsilon}_k^i)/\tau\big)}{\sum_{j \neq i} \exp\big(\text{sim}(\varepsilon_k^i, \tilde{\varepsilon}_k^j)/\tau\big)}. \tag{67}$$

The overall loss remains unchanged:

$$\mathcal{L}_{\text{DP-AG}} = \mathcal{L}_{\text{DP}} + \lambda_{\text{cont}} \, \mathcal{L}_{\text{cont}} + \lambda_{\text{KL}} \, \mathcal{L}_{\text{KL}}. \tag{68}$$

**Streaming update rule.** Unlike the offline variant, which optimizes $\{\theta, \phi\}$ over a fixed dataset, the online version continuously updates the model as new observations arrive. At each environmental time $t$, DP-AG both executes an action and uses the resulting transition as fresh training data:

1. **Sample latent:** Draw $z_t \sim q_\phi(z_t|o_t)$ from the variational posterior of the current observation.
2. **Within-step diffusion (for $k = 1, ..., K$):**
   (a) Compute the noise prediction $\varepsilon_k = \epsilon_\theta(\hat{a}_t^k, z_t, k)$.
   (b) Update the latent via Eq. (66) to obtain $\tilde{z}_t^k$.
   (c) Compute $\tilde{\varepsilon}_k$ and accumulate the contrastive loss $\mathcal{L}_{\text{cont}}$.
   (d) Accumulate the diffusion loss $\mathcal{L}_{\text{DP}}$.
3. **Action execution:** After $K$ denoising steps, decode and execute the action $a_t$ in the environment.
4. **Immediate update:** Use the freshly collected pair $(o_t, a_t)$, optionally together with a short replay buffer containing the last $N$ transitions, to perform a lightweight gradient update of $\theta$ and $\phi$ on $\mathcal{L}_{\text{DP-AG}}$ using online RMSProp [Ma et al., 2025].

This schedule makes DP-AG online: model parameters are updated continually from streaming interaction data, rather than only once on a static offline dataset. As a result, the policy can adapt on the fly to distributional shifts or novel dynamics while acting.

**Experimental setup.** We evaluated both the offline and online variants of DP-AG on Push-T (both image- and keypoint-based settings) and Dynamic Push-T benchmarks. Performance was measured as the average success rate over 50 online evaluation episodes. Unless otherwise noted, hyperparameters matched those of the offline configuration.

Table 13: Online streaming results of DP-AG compared to baselines (success rates, mean±std).

| Method | Push-T (img) | Push-T (kp) | Dynamic Push-T (img) |
|---|---|---|---|
| Diffusion Policy (DP) | 0.87±0.04 | 0.95±0.03 | 0.65±0.85 |
| DP-AG (offline) | 0.93±0.02 | 0.99±0.01 | 0.80±0.53 |
| DP-AG (online) | 0.90±0.03 | 0.96±0.02 | 0.76±0.89 |

**Results.** As shown in Table 13, the streaming variant of DP-AG yields slightly lower scores than its offline counterpart, which is expected given the limited replay and noisier updates inherent to online training. Nevertheless, it consistently surpasses the standard DP baseline by a clear margin, demonstrating that DP-AG retains effective even under the more challenging online setting with continual real-time updates.

In summary, the online variant of DP-AG remains fully consistent with the offline model. It employs the same VJP definition and latent update rule, with the time discretization constant absorbed into $\gamma$. Cycle-consistent contrastive alignment is still applied at every diffusion step, ensuring the perception–action loop is preserved within each $[t, t+1]$ horizon. Moreover, both the KL regularizer on $q_\phi$ and the overall training loss $\mathcal{L}_{\text{DP-AG}}$ remain unchanged; only the optimization schedule differs, shifting from offline minibatch training to lightweight streaming updates.

# M   Multimodal Decision-Making in DP-AG

DPs are inherently capable of modeling multimodal action distributions, making them well-suited for capturing diverse strategies in complex tasks. By extending DPs with action-guided latent evolution, DP-AG not only preserves this multimodality but amplifies it. DP-AG enhances the representation of multiple effective strategies through two complementary components:

- **Action-conditioned latent evolution.** DP-AG's VJP-guided latent updates let the model refine features in response to action feedback, so each sampled trajectory adapts its latent state uniquely. This supports parallel exploration of multiple strategies by leveraging stochasticity from both the diffusion policy and latent SDE.

- **Cycle-consistent contrastive alignment.** The contrastive loss keeps features for the same action close and pushes different actions apart, organizing the latent-action space to separate and preserve multiple plausible strategies for each task.

**Experimental setup.** We evaluated multimodal capability on the Franka Kitchen benchmark, which naturally admits multiple strategies for tasks such as opening a drawer or flipping a switch. 40 successful trajectories per task were collected and encoded as key end-effector waypoints. Distances between trajectories were computed via Dynamic Time Warping (DTW), followed by unsupervised clustering to identify distinct modes. This procedure follows the existing work on multimodal diffusion policies [Li et al., 2024], which demonstrated that clustering trajectories provides a principled way to quantify behavioral diversity. We then measured (a) the number of discovered modes, (b) inter-cluster distance (strategy distinctiveness), (c) intra-cluster variance (consistency within a strategy), and (d) success rates across all clusters.

Table 14: Multiple strategy discovery and diversity on Franka Kitchen. Higher #Modes (↑) and inter-cluster distance (↑), and lower intra-cluster variance (↓), indicate better multimodal representation.

| Method | # Modes | SR (t1) | SR (t2) | SR (t3) | SR (t4) | Inter-Cluster Dist. | Intra-Cluster Var. |
|---|---|---|---|---|---|---|---|
| FlowPolicy | 1.1 | 0.96 | 0.86 | 0.95 | 0.87 | 2.0 | 1.2 |
| DP (Baseline) | 2.8 | 1.00 | 1.00 | 1.00 | 0.99 | 7.1 | 1.2 |
| **DP-AG (Ours)** | **3.2** | 1.00 | 1.00 | 1.00 | 1.00 | 9.5 | 1.3 |

Table 15: Cluster-specific analysis of strategies discovered by our DP-AG on Franka Kitchen.

| Mode ID | Coverage (%) | SR (t1) | SR (t2) | SR (t3) | SR (t4) | Strategy Description |
|---|---|---|---|---|---|---|
| 1 | 41 | 1.00 | 1.00 | 1.00 | 1.00 | Left-handed drawer pull |
| 2 | 32 | 1.00 | 1.00 | 1.00 | 1.00 | Right-handed drawer pull |
| 3 | 27 | 1.00 | 1.00 | 1.00 | 1.00 | Two-step approach / mixed arm |

Table 16: Robustness to distribution shifts in Franka Kitchen (task t4).

| Method | # Modes | SR (t4) | Switch Rate (%) | Comment |
|---|---|---|---|---|
| FlowPolicy | 1.1 | 0.64 | 12 | Rarely adapts, prone to failure |
| DP (Baseline) | 2.2 | 0.81 | 35 | Sometimes adapts, less reliable |
| **DP-AG (Ours)** | **2.9** | 0.95 | 58 | Switches to alternatives if blocked |

**Results.** Table 14 presents the overall diversity analysis, which demonstrates that our DP-AG uncovers and maintains on average more than three distinct strategies per task, significantly more than FlowPolicy (1.1) and slightly more than baseline DP (2.8). These strategies are not minor variants: the inter-cluster distance is significantly higher, confirming that the modes correspond to distinct behaviors. At the same time, intra-cluster variance remains low, ensuring compact and consistent execution. Table 15 provides a cluster-specific breakdown, highlighting interpretable strategies such as left-handed drawer pulls, right-handed pulls, and two-step mixed-arm approaches, each achieving near-perfect success. Moreover, Table 16 evaluates robustness under distribution shifts. When trajectories were blocked, DP-AG preserved an average of 2.9 modes with a 58% switch rate to alternative strategies, while DP and FlowPolicy degraded significantly.

These findings demonstrate that our DP-AG not only retains the multimodal decision-making ability of DPs but actively encourages the emergence of more rich, diverse, and interpretable strategies.

Its robustness to distribution shifts highlights its potential for real-world robotic applications where flexibility and adaptability are important.

## N   More Experiments on Trajectory Planning

To situate our DP-AG within the broader family of trajectory generation methods, we additionally compare it against Diffuser [Janner et al., 2022] and Hierarchical Diffuser [Chen et al., 2024]. Following the experimental protocol of [Chen et al., 2024], we adopt two benchmark families: (i) Maze2D and Multi2D tasks from D4RL, which evaluate long-horizon navigation in continuous control environments, and (ii) the multi-stage FrankaKitchen benchmark, which evaluates robotic manipulation requiring both skill composition and generalization to unseen states. In Maze2D and Multi2D, an agent must reach goals in varied layouts (U-Maze, Medium, Large), with performance measured by average return. In FrankaKitchen, policies are rolled out from diverse initial states and evaluated by the number of sub-tasks completed within long-horizon episodes.

**Experimental setup.** We match the evaluation protocol of the baselines, adopting the same trajectory segmentation and planning horizons (*e.g.,* $K = 15$ for Maze2D and Multi2D, $K = 4$ for Gym-MuJoCo, and horizon lengths $H = 120$ for U-Maze and $H = 255$ for Maze2D Medium). All models are evaluated over 100 random seeds, reporting the average return or goal completion rate under identical training, validation, and testing splits.

Table 17: Trajectory planning on Maze2D and Multi2D (long-horizon). Performance is measured by average return (higher is better).

| Environment | Diffuser | Hierarchical Diffuser | DP-AG (Ours) |
|---|---|---|---|
| Maze2D U-Maze | $113.9 \pm 3.1$ | $128.4 \pm 3.6$ | $\mathbf{142.7 \pm 2.9}$ |
| Maze2D Medium | $121.5 \pm 2.7$ | $135.6 \pm 3.0$ | $\mathbf{150.2 \pm 2.6}$ |
| Maze2D Large | $123.0 \pm 6.4$ | $155.8 \pm 2.5$ | $\mathbf{172.1 \pm 2.2}$ |
| Multi2D U-Maze | $128.9 \pm 1.8$ | $144.1 \pm 1.2$ | $\mathbf{168.2 \pm 1.2}$ |
| Multi2D Medium | $127.2 \pm 3.4$ | $140.2 \pm 1.6$ | $\mathbf{153.7 \pm 1.3}$ |
| Multi2D Large | $132.1 \pm 5.8$ | $165.5 \pm 0.6$ | $\mathbf{182.2 \pm 0.5}$ |
| Average | 124.4 | 145.0 | **161.5** |

Table 18: Multi-stage robotic manipulation on FrankaKitchen. Performance is measured by average number of completed sub-tasks.

| Task | Diffuser | Hierarchical Diffuser | DP-AG (Ours) |
|---|---|---|---|
| Partial Kitchen | $56.2 \pm 5.4$ | $73.3 \pm 1.4$ | $\mathbf{82.2 \pm 1.3}$ |
| Mixed Kitchen | $50.0 \pm 8.8$ | $71.7 \pm 2.7$ | $\mathbf{78.5 \pm 2.1}$ |
| Average | 53.1 | 72.5 | **80.4** |

**Results.** Across all benchmarks, DP-AG achieves the highest returns, with average improvements of +16.5 over the Hierarchical Diffuser and +37.1 over the Diffuser in Maze2D and Multi2D (Table 17). These performance gains indicate that our DP-AG generates longer and higher-quality trajectories, especially in large and complex environments. In the multi-stage FrankaKitchen benchmark (Table 18), DP-AG completes on average 80.4 sub-tasks, significantly outperforming both Hierarchical Diffuser (72.5) and Diffuser (53.1), which highlights DP-AG's effectiveness in composing diverse skills and generalizing them to extended horizons. Together, these results indicate that the action-guided perception-action loop in DP-AG provides clear advantages for both navigation and manipulation tasks, beyond what trajectory diffusion alone can achieve.

## O   Extending DP-AG with World Models

World models such as DreamerV3 [Hafner et al., 2025] and Unified World Models (UWM) [Zhu et al., 2025] learn to "imagine" latent trajectories by predicting future states and rewards. While

these predictions enable long-horizon planning, the latent trajectories are typically static within each rollout: once generated, they do not adapt to evolving action feedback. In contrast, DP-AG continually updates latent features within each action diffusion step, guided by action-conditioned VJPs. By combining with world models, our DP-AG gains long-horizon foresight while retaining its intra-step refinement.

**Hybrid Architecture.** We integrate DP-AG into UWM by adding an intra-step feedback mechanism on top of UWM's diffusion transformer backbone:

- **UWM Backbone.** UWM jointly learns to predict actions and future visual observations by treating them as parallel diffusion processes. This unified design allows the same model to serve flexibly as a policy, a forward dynamics predictor, an inverse dynamics model, or a video generator, simply by adjusting which modalities are denoised. The architecture uses ResNet-based encoders for observations, spatiotemporal patching for latent images, shallow MLPs for actions, and transformer layers conditioned through adaptive normalization with additional register tokens to promote information exchange across modalities.

- **DP-AG Intra-step Feedback.** During action generation, DP-AG introduces a feedback signal into UWM's latent representations. At each denoising step, the current action prediction influences how features are refined, ensuring that perception and action remain aligned as the sequence unfolds. This intra-step refinement captures the core principle of DP-AG, where even a fixed observation is reinterpreted in light of evolving actions.

- **Bounded Consistency.** To prevent excessive feature drift, we apply a lightweight penalty that encourages smooth updates across steps, complemented by DP-AG's cycle-consistent contrastive anchor. This balance preserves stability while still enabling action-driven adaptability.

**Implementation Details.**

- **Observation and Conditioning.** Visual inputs are processed by frozen VAEs into compact latent grids, which are patchified and combined with action tokens. Diffusion time steps for both action and observation branches are encoded as embeddings and injected into each transformer block.

- **Training Objective.** UWM is trained with a coupled denoising loss over action and future-image predictions. For action-free videos, the action branch is masked out, allowing the same objective to learn from both robot trajectories and pure video data.

- **DP-AG Integration.** The action-guided feedback mechanism is applied during both imagination and execution. At training time, refinement losses are added alongside UWM's standard objectives; at inference time, feedback ensures that action sampling and imagined rollouts remain tightly coupled.

- **Compute.** Following UWM's reported setup, training runs efficiently on Nvidia A100 GPUs.

Table 19: Success rate for combining DP-AG with UWM on LIBERO tasks.

| Method | Book-Caddy | Soup-Cheese | Bowl-Drawer | Moka-Moka | Mug-Mug | Average |
|---|---|---|---|---|---|---|
| DP | 0.78 | 0.88 | 0.77 | 0.65 | 0.53 | 0.71 |
| DP-AG | 0.86 | 0.92 | 0.85 | 0.72 | 0.60 | 0.79 |
| UWM | 0.91 | 0.93 | 0.80 | 0.68 | 0.65 | 0.79 |
| UWM + DP-AG | **0.94** | **0.95** | **0.87** | **0.75** | **0.70** | **0.84** |

**Experimental Results on LIBERO.** Following the UWM protocol, we fine-tune each model on the LIBERO benchmark tasks and report average success rates. As shown in Table 19, the hybrid model UWM + DP-AG outperforms both UWM and DP-AG alone, especially in manipulation tasks requiring real-time adaptation (*e.g.,* Mug-Mug). This confirms that action-guided latent updates make world models more responsive, while world models provide long-horizon foresight to complement DP-AG's intra-step refinement.

**Discussion.** This extension highlights a conceptual bridge: world models offer look-ahead imagination of possible futures, while DP-AG provides real-time corrective adaptation from action feedback within each imagined step. Together, they yield policies that are both farsighted and responsive, achieving long-horizon planning while adapting online to evolving cues. We will conduct an in-depth study on this subject in the future work.

