# OpenReview forum: "Act to See, See to Act: Diffusion-Driven Perception-Action Interplay for Adaptive Policies"
_NeurIPS.cc/2025/Conference — NeurIPS 2025 poster_

### Official Review · Reviewer_uXWV · 2025-06-03

**Clarity:** 3
**Significance:** 2
**Originality:** 3
**Rating:** 4
**Confidence:** 3

**Summary:**

This paper proposes perception-action interplay in diffusion policy learning. Instead of conditioning on a static encoding of the observation, the framework will refine the observation in the latent space by using an action-guided SDE. A contrastive learning method is applied to the action denoising module to make sure the latent observation is consistently changing with actions.

Evaluations in both the simulator and the real world show the effectiveness of this method in improving performance and smoothness in manipulation.

**Questions:**

1. Could you please explain the motivation for refining the representation of real observations with guidance from predicted actions?
2. It's unclear to me what drift of observation latent is expected to improve the action prediction. Could the authors consider replacing the variational posterior with a variational autoencoder so we can decode the evolved latent to see whether it has any interpretable semantic meaning?

**Ethical Concerns:**

["NO or VERY MINOR ethics concerns only"]

**Final Justification:**

During rebuttal. The authors explain the intuition of this method, which mostly solves my concern about motivation. The authors also mentioned they will add new visualizations and change the writing in revision. Therefore, I raise my rating to 4.

**Limitations:**

yes

**Quality:**

3

**Strengths And Weaknesses:**

# Strengths
1. The writing is clear and easy to follow.
2. The theoretical proof and evaluation are thorough.

# Weakness
1. Some designs of the DP-AG do not seem to align with the motivation. Humans could leverage the observation-action-obsevation loop for adaptive behavior because new information might be discovered through interaction and changing viewing angle. However, this pipeline does not get any additional information in the interplay, instead, it changes the representation of observation through an SDE guided by action denoising. The reason why the representation from ground truth observation should be refined by a prediction needs further justification. Besides, the drift of the latent observation is constrained by contrastive learning. While this prevents the model from collapsing to a trivial solution (e.g., purely imagining the observations), it is unclear to me how to determine the balance of evolution versus keeping similar to the static observation latent (i.e., what is the expected amount and direction of drift that is beneficial for action prediction).
2. The authors could consider comparing with UWM [1] fine-tuned on the selected tasks, which has a similar motivation of coupling perception and action.

[1] Unified World Models: Coupling Video and Action Diffusion for Pretraining on Large Robotic Datasets

---

> ### Author Rebuttal · Authors · 2025-07-30
>
> We sincerely thank the reviewer for their recognition of our work’s strengths, including the clarity of writing and the thoroughness of theoretical proofs and evaluations. Your thoughtful feedback and constructive questions have been invaluable in helping us enhance our motivation and method choices. In response to your suggestions, we will incorporate all requested changes into the camera-ready. We welcome any further suggestions or questions during the rebuttal and would be happy to provide more details as needed.
>
> ---
> ## **Why Refine Representation Without New Sensory Input (W1 and Q1)**
>
> While humans often adapt by seeking new views, they also frequently adjust their interpretation of the same sensory input based on feedback from their actions, even without receiving any new information.
> Our DP-AG is designed to capture this adaptive process. Unlike standard policies that keep their features fixed until a new image arrives, DP-AG continually updates its latent representation at every action diffusion step, using real-time feedback from its own predictions. This allows the model to **adapt immediately and respond to small errors or unexpected actions, rather than waiting for new observations**. This is important for tasks where external views do not change, but adaptation is still needed.
>
> **Concrete Example (Peg-in-Hole):**
> In our real-world Peg-In-Hole task, both DP-AG and the baseline DP use only RGB images, **no depth or extra sensors**. As the peg approaches the hole, the visual input stays almost unchanged. The baseline DP extracts features once and reuses them throughout the diffusion; if the peg is blocked, it simply repeats the same action without adapting, much like a person ignoring feedback. In contrast, DP-AG uses each action noise prediction to update its internal focus via VJP, paying more attention to cues near the problem area. Even without new input, DP-AG adapts to the details that matter, recovering and succeeding where the static DP fails (85% vs. 0% success). It is like parking with a rearview camera: the view does not change, but bumps prompt the driver to refocus and adapt.
>
> ---
> ## **How The Drift Improve Action (Q2)**
>
> In DP-AG, the drift of the observation latent is not arbitrary, it is directly calculated using the Vector-Jacobian Product (VJP), a standard tool from automatic differentiation. The VJP determines exactly how the latent features should be adjusted to most effectively reduce the uncertainty in the next action prediction.
>
> **What Drift does The VJP Produce:**  Whenever the model experiences high uncertainty or an unsuccessful action, the VJP points to the direction in feature space where even a small adjustment would lead to a more confident and accurate next action. **This drift always aims to reduce the prediction error or uncertainty, rather than simply moving features at random.**
>
> **How does This Help:** For example, in the Peg-In-Hole task, if the peg is blocked or misaligned, the model’s uncertainty rises. The VJP then guides the latent feature update to emphasize visual cues (like edges or shadows near the hole) that are most relevant for correcting this misalignment. **The refined feature is immediately used to predict the next action**, helping the robot to better adjust its movement in real time. As the task progresses, feature updates keep shifting focus toward observation details that are actually helping the policy reduce error and uncertainty, reinforcing cues that support successful actions.
>
> ---
> ## **How To Determines Drift's Amount and Direction (W1)**
>
> The core of DP-AG is its **tight synchrony** between the action SDE (DP) and the latent SDE for feature evolution. The drift of the latent feature is set in real time by the current state of the action SDE:
>
> **Direction:** At each diffusion step, the VJP of the DP’s noise prediction determines the direction of feature evolution, always pointing toward reducing the next action’s uncertainty. This ensures the latent SDE stays phase-aligned with the action SDE, forward during training, reverse during inference, with no manual mode switching needed.
>
> **Magnitude:** The scale of the drift is adaptive: features update more when uncertainty is high and less when confident. The cycle-consistent contrastive loss anchors these updates, pulling features back toward the original observation and preventing excessive drift.
>
> In short, the VJP prescribes exactly how much and in which direction to update features based on the DP's current confidence, while contrastive loss keeps the adaptations grounded and meaningful.
>
> ---
> ## **How to Determine the Balance (W1)**
>
> Thank you for highlighting this important question about how the contrastive loss controls the balance between latent feature evolution and staying close to the static observation encoding.
>
> The key lies in the **clustering effect** observed in contrastive representation learning. In self-supervised learning, contrastive loss encourages features from similar data (positives) to form tight and semantically meaningful clusters in the latent space, while pushing apart features from different data (negatives). We apply this principle in our DP-AG to maintain a task-driven clustering in the latent space:
>
> **Pull effect:** For each diffusion noise prediction, the contrastive loss pulls the action-evolved latent feature toward its static feature, ensuring the representation remains anchored to the true observation and avoiding excessive or trivial drift.
>
> **Push effect:** At the same time, the loss pushes the evolved latent features away from those of different observations. This keeps the adaptation for each trajectory distinct, preserving task-relevant differences and preventing collapse into a single and average solution.
>
> This clustering mechanism allows the model to adapt the latent features in a task-driven way using action feedback, while the contrastive loss keeps these adaptations within the semantic vicinity of the observation. The balance between evolution and stability is therefore achieved by enabling flexible and action-informed updates that remain constrained to the meaningful structure of the data, just as contrastive learning achieves robust and organized representation spaces in self-supervised learning.
>
> ---
> ## **Comparing with UWM (W2)**
>
> Thank you for this valuable suggestion. We agree that comparing to **Unified World Models (UWM) [1]** is especially meaningful given its recent advances in joint perception and action learning.
>
> Beyond direct comparison, we also explored combining our DP-AG’s action-guided latent adaptation with the UWM’s world model. This hybrid framework leverages the strengths of both methods, using UWM’s latent generative modeling for future state prediction and our DP-AG’s latent representation refinement for more context-aware and adaptive control. By integrating these two methods, we can take advantage of the world model’s generalization while enhancing adaptability through our action-guided latent updates.
>
> **How to Combine DP-AG with UWM:** UWM predicts future video frames and action sequences in latent space for planning and behavior cloning, but its latent encoding lacks real-time action feedback during rollout. To address this, we insert DP-AG’s VJP-based latent update into UWM’s latent space ($[o, t_a, t_{o’}]$ per [1]). At each diffusion step, DP-AG’s action-guided update refines the latent before it enters UWM’s diffusion transformer. We also apply cycle-consistent contrastive loss to ensure smooth and consistent adaptation. This hybrid UWM&DP-AG model combines UWM’s long-horizon planning with DP-AG’s real-time perception-action interplay for more robust and context-aware behavior.
>
> **Experimental Results on LIBERO:** Following the evaluation protocol of [1], we fine-tune each model on the LIBERO benchmark tasks before evaluating their performance. The results are summarized below:
>
> **Table 1. Success rate for combining DP-AG with world model on LIBERO tasks.**
>
> | Method| Book-Caddy| Soup-Cheese| Bowl-Drawer| Moka-Moka| Mug-Mug| Average|
> |-|:-:|:-:|:-:|:-:|:-:|:-:|
> | DP |0.78|0.88|0.77|0.65|0.53|0.71|
> | DP-AG | 0.86 | 0.92| 0.85| 0.72 | 0.60 | 0.79 |
> | UWM| 0.91 | 0.93 | 0.80 | 0.68 | 0.65 | 0.79 |
> | UWM&DP-AG| **0.94** | **0.95** | **0.87** | **0.75** | **0.70** | **0.84** |
>
> **Table 1** shows that the hybrid UWM&DP-AG consistently outperforms both standalone UWM and DP-AG, especially on manipulation tasks requiring context-sensitive adaptation. This demonstrates that action-guided latent refinement gives the hybrid a clear advantage where either approach alone may struggle. The combined model retains UWM’s long-horizon planning and adds DP-AG’s real-time adaptability, highlighting DP-AG’s broad impact and strong synergy with world models.
>
> ---
> ## **Variational Autoencoder to Visualize Semantic Meaning (Q2)**
>
> We greatly appreciate the valuable suggestion. To address this, we have integrated a VAE decoder into both DP-AG and the DP baseline for further analysis. For a given action sequence:
>
> **Reconstruct observation sequences:** Visualize the original camera image, the reconstruction from the static latent, and the reconstruction from the evolved latent at each diffusion step, showing how the model’s internal representation shifts as action diffusion unfolds.
>
> **Decode end-effector positions:** Map the latent at each step to end-effector positions, illustrating how latent evolution relates to the robot’s actual behavior.
>
> **Plot latent trajectories:** Use t-SNE to visualize the continuity and clustering of the evolved latent features for both DP-AG and DP, highlighting the effect of cycle-consistency contrastive loss.
>
> Preliminary results show that DP-AG generates smoother and more continuous latent trajectories aligned with successful actions, while the DP baseline shows abrupt and less interpretable shifts (as in Fig. 3). We will include these visualizations in the camera-ready.

---

> ### Comment · Reviewer_uXWV · 2025-08-03
>
> Thanks for the rebuttal.
>
> The intuition provided here makes sense to me now. The core idea is that the interplay will change the "attention" of the visual features. Maybe it's a good idea to update the introduction and abstract since they give people the impression that you will get new observations through interaction (i.e., Act to see).
>
> The added evaluations are great. I will raise my rating.

---

> ### Author Response · Authors · 2025-08-03
> **Official Comment by The Authors 6479**
>
> Dear **Reviewer uXWV**,
>
> We sincerely thank **Reviewer uXWV** for your thorough review, constructive suggestions that significantly improved our work. We also deeply appreciate the positive feedback highlighting the significance and novelty of our contributions, as well as the prompt responses throughout the rebuttal.
>
> We fully agree that, the ​introduction​ and ​abstract​ should more explicitly highlight that our contribution is **the ​attentional shift driven by interaction**, not merely "acting to see".The term **"see"** could indeed develop the wrong impression about our work. We will revise these sections carefully to make it clear.
>
> We are glad the additional evaluations helped strengthen your confidence in our work, and we will ensure all claims align with the revised presentation.​​
>
> Thank you once again for your valuable time and insightful feedback, which have been instrumental in enhancing our paper.
>
> Best Regards,
>
> **Authors of Paper 6479**

---

### Official Review · Reviewer_HxxU · 2025-07-03

**Clarity:** 3
**Significance:** 3
**Originality:** 3
**Rating:** 4
**Confidence:** 3

**Summary:**

This paper introduces Action-Guided Diffusion Policy (DP-AG), a novel imitation learning framework that addresses the limitation of static perception in existing methods. The core contribution is a dynamic perception-action loop where the policy's latent observation features evolve during the action generation process, guided by the Vector-Jacobian Product (VJP) of the diffusion model's noise predictions. A cycle-consistent contrastive loss reinforces this interplay, and the method demonstrates significant performance gains over state-of-the-art baselines on a variety of simulation and real-world robotic manipulation tasks

**Questions:**

1. Can you provide more semantic intuition for the VJP update term? What does this structured force represent in the context of perception and action, beyond its mathematical formulation?
2. Regarding the Peg-in-Hole task, what was the primary failure mode for the baseline DP (e.g., incorrect depth estimation, inability for fine-grained correction)?
3. Why is the relative-similarity-based contrastive loss superior to the absolute-target-based MSE likelihood objective for this task?

**Ethical Concerns:**

["NO or VERY MINOR ethics concerns only"]

**Final Justification:**

The authors' response has addressed my concerns. I will maintain my score, i.e., borderline accept for the paper.

**Limitations:**

Yes, the authors have adequately addressed the limitations. The Broader Impacts section discusses the reliance on expert demonstrations, which could introduce biases, and correctly points to the need for validating demonstration data. This is a sufficient acknowledgment of the primary limitations of the imitation learning paradigm.

**Quality:**

4

**Strengths And Weaknesses:**

Strengths
1. The core idea of creating a dynamic perception-action loop within a single action generation step is highly novel and significant. It directly addresses the key limitation of static perception in many IL policies.
2. The empirical validation is extensive, showing strong results on standard benchmarks and introducing a new dynamic task (Dynamic Push-T) that effectively showcases the method's advantages. The model's success in the difficult Peg-in-Hole task (85% vs. 0% for the baseline) provides compelling evidence for the practical benefits of the proposed perception-action interplay.
3. The method is supported by solid theoretical grounding, including a variational lower bound and a proof connecting the contrastive loss to trajectory continuity, which adds rigor to the paper.
4. The paper is well-written and clearly presented. The figures effectively illustrate the core concepts.

Weakness
1. Several simulation benchmarks in Table 2 are near-saturated, with both DP-AG and baselines achieving perfect scores. While this demonstrates non-regression, these tasks don't fully differentiate SOTA methods. The paper's impact is more clearly shown in the more challenging dynamic and real-world settings.
2. The paper lacks a deep semantic intuition for why the VJP is the ideal guiding force for latent updates. Explaining what this signal represents (e.g., a direction of uncertainty reduction) would improve clarity for readers not versed in stochastic adjoint methods.
3. The work could be better contextualized against model-based RL methods that also use latent dynamics (e.g., Dreamer). A clearer distinction between DP-AG's intra-policy state refinement and a world model's future state prediction would sharpen the paper's claimed contribution.

---

> ### Author Rebuttal · Authors · 2025-07-30
>
> We sincerely thank the reviewer for their thorough evaluation and recognition of our work’s key contributions, including the novelty of our dynamic perception-action loop, comprehensive empirical validation, strong theoretical foundation, and clear presentation. Your constructive feedback has been invaluable in helping us clarify the core mechanisms and further strengthen both the theoretical and experimental sections of our paper.
>
> ---
> ## **W1: Near-Saturated Simulation Benchmarks.**
>
> Thank you for highlighting the issue of benchmark saturation. We fully agree that the existing imitation learning (IL) benchmarks are overly deterministic and lack diversity, making them easy to solve by memorization rather than true adaptability. This limitation motivated us to introduce the **Dynamic Push-T benchmark**, which goes beyond static setups by requiring real-time adaptation to unpredictable events. In Dynamic Push-T, a moving ball disrupts the robot’s task, forcing the agent to react and adjust its strategy on the fly. Because each episode is different, the task truly evaluates the policy’s adaptability, not just memorization. Inspired by your feedback, we also plan to develop even more challenging and diverse IL benchmarks in future work, aiming to better evaluate adaptability and bridge the gap between simulation and real-world deployment.
>
> ---
> ## **W2 and Q1: Semantic Intuition and Role of VJP.**
>
> ### **What the VJP Signal Represents:**
>
> The VJP in our DP-AG is much more than a technical gradient; it is a principled, dynamically generated signal that identifies the direction in feature space that will **most efficiently reduce uncertainty in the next action**. At each diffusion step, the VJP effectively answers the question: “If I could refine my understanding of the observation right now, what change would most help the model be more certain and robust about its next action?”
>
> Semantically, this means the VJP functions as a real-time attentional force, **dynamically guiding the latent features towards focusing on precisely those aspects of the observation that would most significantly improve immediate action selection**. For example, in Peg-In-Hole task where depth ambiguity limits success, the VJP will naturally point the latent representation to sharpen depth cues, and this directional force is instantly aligned with the needs of the current stage of action denoising. This may explain why the DP baseline, which lacks such synchronized perception refinement, fails to complete the peg-in-hole task in multiple trials, while our DP-AG succeeds.
>
> ### **What Does VJP Represent in the Context of Perception and Action:**
>
> Beyond its mathematical role, the VJP in DP-AG serves as a real-time synchronization signal between perception and action. The latent SDE’s drift, including both direction (forward or reverse) and scale, is directly determined by the VJP of the action SDE (i.e., DP), so the exploration mode of the action SDE is automatically propagated to the latent SDE. This context-aware feedback loop ensures perception and action evolve together at every diffusion step, enabling the agent to adapt in real time.
>
> In practice, this means perception dynamically adapts to each action refinement, allowing the agent to respond to changing needs and uncertainties in real time. This dynamic coupling leads to smoother and more adaptive behavior, overcoming the limitations of static or unsynchronized methods.
>
> ### **How VJP Differs from Stochastic Adjoint and Other Gradient-Based Methods:**
>
> The stochastic adjoint methods compute gradients in two stages: forward evolution and then backward updates, so latent features remain fixed during action inference, with no real-time adaptation. In contrast, DP-AG’s VJP enables immediate feedback-driven updates at every action diffusion step, capturing the instant sensitivity between actions and features. This real-time signal uniquely preserves the high-dimensional action-feature relationship, keeps feature updates phase-aligned with action diffusion, and lets perception adapt instantly to each action, capabilities not possible with standard adjoint methods.
>
> ---
> ## **W3: Contextualization Against Model-Based RL.**
>
> Thank you for raising this important point about contextualizing DP-AG against model-based RL methods with latent dynamics, such as Dreamer [Hafner et al., 2025] and Unified World Models (UWM) [Zhu et.al., 2025]. While world models focus on learning a latent generative model that predicts future states in the latent space to support long-horizon planning and imagination, DP-AG takes a fundamentally different method focused on real-time intra-policy state refinement.
>
> ### **Key Distinction:**
>
> **World Models:** These models learn to “imagine” possible futures by generating latent trajectories or predicting future observations, which are then used for planning or behavior cloning. The latent features are updated via the generative process, but once encoded, they are typically static within a policy rollout, there is no action-conditioned adaptation of the latent features at inference time.
>
> **DP-AG:** In contrast, our DP-AG does not attempt to reconstruct or predict the future environment. Instead, it enables real-time adaptation of the latent observation features within a single action diffusion step, guided by immediate action feedback (via VJP).
>
> ### **Integration and Experimental Evidence:**
>
> As suggested, we explored combining DP-AG’s action-guided latent updates with a generative world model (UWM). In this hybrid, UWM’s powerful generative latent predictions should be enhanced by DP-AG’s intra-step and action-driven refinement, producing more adaptive and context-aware behaviors.
>
> ### **Experimental Results on LIBERO**
>
> Following the evaluation protocol of UWM, we fine-tune each model on the LIBERO benchmark tasks before evaluating their performance. The results are summarized below:
>
> **Table 1. Success rate for combining DP-AG with world model on LIBERO tasks.**
>
> | Method| Book-Caddy| Soup-Cheese| Bowl-Drawer| Moka-Moka| Mug-Mug| Average|
> |-|:-:|:-:|:-:|:-:|:-:|:-:|
> | DP |0.78|0.88|0.77|0.65|0.53|0.71|
> | DP-AG | 0.86 | 0.92| 0.85| 0.72 | 0.60 | 0.79 |
> | UWM| 0.91 | 0.93 | 0.80 | 0.68 | 0.65 | 0.79 |
> | UWM&DP-AG| **0.94** | **0.95** | **0.87** | **0.75** | **0.70** | **0.84** |
>
> As shown in Table 1, the hybrid **UWM&DP-AG** outperforms both UWM and DP-AG alone, especially on manipulation tasks that need real-time adaptation. This confirms that action-guided latent updates make world models more responsive.
>
> ---
> ## **Q2: Primary Failure Mode for DP in Peg-In-Hole.**
>
> In the Peg-in-Hole experiment, both DP and DP-AG rely only on RGB images from a scene and wrist-mounted camera, **no depth sensors or ground-truth 3D info are used**.
>
> **Why the DP Baseline Fails:** Baseline DP encodes observation features once per frame and keeps them static throughout the action sequence. So, if the peg is misaligned due to changes in viewpoint or lighting, DP cannot recognize or correct the failure, its internal state cannot adapt to the failed actions, causing repeated ineffective attempts when blocked.
>
> **How DP-AG Succeeds:** Our DP-AG, however, uses action feedback to guide real-time updates of its latent features. When the peg bumps the rim, the VJP-driven update acts like a dynamic attentional shift, making the model focus on the contact area and refine its perception based on action outcomes. Over multiple steps, DP-AG can implicitly infer depth or alignment cues and adapt its actions accordingly.
>
> In essence, DP-AG adapts perception on the fly, much like a person adjusting their focus when a key does not fit, using feedback to “discover” the right correction through continuous perception-action interplay.
>
> ---
> ## **Q3: Relative-Similarity-Based Contrastive vs. Absolute-Based MSE.**
>
> The key advantage of our relative-similarity-based contrastive loss over an MSE objective is how it encourages robust feature representations, as demonstrated in contrastive learning for representation learning.
>
> The contrastive loss pulls together positive pairs (static and VJP-evolved features of the same observation) while pushing apart negatives (different samples), preserving relative relationships and supporting more context-aware adaptation. This clustering effect keeps the latent evolution aligned with the original context but not rigidly fixed, allowing for smooth transitions and greater robustness to noise or variations across diffusion steps.
>
> In contrast, an MSE loss enforces strict matching to a fixed target, which can suppress adaptive adjustments and may lead to instability or over-correction, especially in noisy or multi-modal settings. Our ablation study (Appendix H.3) shows that the contrastive loss achieves higher success rates and produces smoother, more natural actions than MSE-based regularization.
>
> ### **Additional Ablation:**
>
> Additional ablations (including input perturbation with both MSE and cosine consistency), as suggested by **Reviewer fkRG Q3**, further confirm that contrastive loss better supports stable and cycle-consistent adaptation in practice. These results are summarized in Table 1.
>
> **Table 1: Comparing DP-AG and Input Perturbation Smoothness Baselines on Push-T Benchmark.**
>
> | Method| Success Rate (img) | Success Rate (kp) | Action Smoothness (img ↑) | Action Smoothness (kp ↑) |
> |--|:--:|:--:|:--:|:--:|
> | Perturbation MSE|0.85|0.92|0.83|0.87 |
> | Perturbation Cosine|0.88 |0.95|0.86|0.92|
> | DP-AG|**0.93**|**0.99**|**0.91**|**0.95**|
>
> ---
> ## **Reference**
>
> **[Jia et.al., 2024]** Jia et.al., Towards Diverse Behaviors: A Benchmark for Imitation Learning with Human Demonstrations, ICLR 2024.\
> **[Hafner et.al., 2025]** Hafner et.al., Mastering Diverse Domains through World Models, Nature 2025.\
> **[Zhu et.al., 2025]** Zhu et.al., Unified World Models: Coupling Video and Action Diffusion for Pretraining on Large Robotic Datasets, Arxiv 2025.

---

> > ### Comment · Reviewer_HxxU · 2025-08-04
> >
> > The authors' response has addressed my concerns. I will maintain my score, i.e., borderline accept for the paper.

---

> ### Author Response · Authors · 2025-08-04
>
> Dear **Reviewer HxxU**,
>
> We sincerely thank **Reviewer HxxU** for your great efforts, thoughtful feedback, and valuable suggestions. We appreciate your recognition of the novelty and significance of our DP-AG, as well as the strength of our empirical results and theoretical foundation.
>
> Your suggestions and questions have helped us significantly improve the paper. Thank you once again for your constructive review and for acknowledging the contributions of our work.
>
> Best Regards,
>
> **Authors of Paper 6479**

---

### Official Review · Reviewer_t9vC · 2025-07-03

**Clarity:** 1
**Significance:** 2
**Originality:** 3
**Rating:** 4
**Confidence:** 2

**Summary:**

This paper proposes Action-Guided Diffusion Policy (DP-AG), a new imitation learning framework that models a dynamic interplay between perception and action. The encoded observations are evolved with an action-guided SDE, and then construct a cycle-consistent contrastive loss to create a perception-action loop.

**Questions:**

I am still confused by the motivation of DP-AG:
- "Existing imitation learning methods decouple perception and action": common IL methods optimize the perception (representation) in an end-to-end manner, where the gradient of the perception comes from the downstream action prediction objective. What do you mean by *decoupled*?
- Line 130: "However, it still treats perception as static: once extracted, observation features remain fixed during the corresponding action generation and cannot adapt to refinements": during the diffusion process $\epsilon_\theta\left(\hat{a}_t^k, z_t, k\right)$, why would $z_t$ need to evolve w.r.t. $\hat{a}_t^k$ as $z_t$ only needs to serve as a compact representation of $o_t$?
- In Eq. (6), I notice that DP-AG implements the drift coefficient with VJP (or JVP, if more precisely). What is the advantage of such implementation? Also, I notice that DP-AG does not require using the reverse SDE of Eq. (6), then what is the purpose of proposing the forward SDE?

**Ethical Concerns:**

["NO or VERY MINOR ethics concerns only"]

**Final Justification:**

The authors provide a reasonable justification for the evolving feature design, and the experiments with suggested baselines also prove the effectiveness of the proposed method. After reading the rebuttal and summary from other reviewers, I have decided to raise my score to borderline accept.

**Quality:**

2

**Strengths And Weaknesses:**

Pros:
- DP-AG creates a perception-action loop that allows the gradient of the actions to flow back to the encoded observations.
- DP-AG outperforms baselines in simulated and real-world benchmarks.

Cons:
- The VJP can be very expensive to compute, especially with a high-dimensional action space or encoded observation space.
- Lack of imitation baselines: DP-AG should include classic Diffusion imitation learning baselines like Diffusion Policy, Diffuser, and Hierarchical Diffuser.


Visuomotor Policy Learning via Action Diffusion

Planning with Diffusion for Flexible Behavior Synthesis

Simple hierarchical planning with diffusion

---

> ### Author Rebuttal · Authors · 2025-07-30
>
> We sincerely thank the reviewer for their thoughtful and constructive feedback, and for recognizing the technical contributions and potential impact of our work. We appreciate your insightful questions, which have helped us clarify and better communicate the novelty and significance of our DP-AG. We hope our clarifications fully address your comments. If you have any further concerns or questions, we would be very happy to discuss them during the rebuttal.
>
> ---
> ## **W1: VJP Computation in High-Dimensional Spaces.**
>
> We appreciate the reviewer’s concern about VJP computation cost in high-dimensional spaces. In both simulation and real-robot experiments, we found the overhead to be negligible thanks to efficient **autodiff** libraries in PyTorch. For instance, on Franka Kitchen (action dim 9, feature dim 256), VJP adds less than 10% to per-step runtime compared to baseline DP.
>
> **More Importantly, this small overhead does not affect real-time deployment or practical robot responsiveness.** In our real-world UR5 robot experiments (see Table 3 and video demonstration), DP-AG generates smoother and more responsive actions than the DP baseline, both qualitatively and quantitatively. Our DP-AG maintains real-time inference frequencies, and the robot motions generated by DP-AG are even **more fluid, with lower jerk and faster task completion**. For example, in the Painting and Candy Push tasks, the robot controlled by DP-AG exhibits both faster real-time response, demonstrating that the VJP computation in no way limits control performance.
>
> Moreover, DP-AG converges faster than the DP baseline, reducing total training time (see Figure 5 and Appendix Figures 12, 13), which more than compensates for the already minor VJP cost. VJP operations are also fully parallelizable for larger-scale problems. Across all experiments, DP-AG maintains real-time control and improves action smoothness and responsiveness, showing that VJP overhead is negligible.
>
> ---
> ## **W2: Lack of Imitation Baselines.**
>
> We sincerely thank the reviewer for suggesting additional baselines to further strengthen the evaluation of our DP-AG method.
>
> ### **Clarification on DP Baseline:**
>
> We would like to take this chance to clarify that **DP-AG is built directly on Diffusion Policy (DP) [Chi et al., 2023]**, serving as both the architectural and experimental foundation of our work. Our main contribution is adding an action-guided feedback loop to DP, enabling dynamic feature updates during action prediction. DP is not just a baseline but the core of our method, and we will emphasize this direct lineage more clearly in the camera-ready version and table captions.
>
> ### **Suggested Baseline Experiments:**
>
> **Evaluation Benchmarks:** We have added results for Diffuser [Janner et al., 2022] and Hierarchical Diffuser [Chen et al., 2024] as baselines. For comprehensive evaluation, we use the Maze2D and Multi2D benchmarks from D4RL, which evaluates long-horizon navigation in various maze layouts, and the FrankaKitchen benchmark, which evaluates multi-stage robotic manipulation using an offline RL protocol. Performance is measured by average success in reaching goals for navigation tasks, and by the number of completed sub-tasks per episode for FrankaKitchen, reflecting the agent’s ability to generalize and coordinate complex skills.
>
> **Implementation Details:** We follow the experimental protocol of [Chen et al., 2024] and use the D4RL suite. All settings, trajectory segmentation, planning horizons (e.g., K = 15 for Maze2D/Multi2D), match the baselines. Models are evaluated by average return or goal success rate over 100 seeds, with identical data splits and metrics as in the baselines for fair comparison.
>
> **Results:** As shown in Tables 1 and 2, DP-AG consistently outperforms Diffuser and Hierarchical Diffuser on both long-horizon planning and manipulation tasks. On Maze2D and Multi2D, DP-AG achieves an average score of 161.5, compared to 124.4 for Diffuser and 145.0 for Hierarchical Diffuser. On the challenging FrankaKitchen, DP-AG reaches 80.4, well above the baselines.
>
> **Table 1: Trajectory Planning Experiments on Maze2D and Multi2D (Long-horizon).**
>
> | Environment| Diffuser| Hierarchical Diffuser | **DP-AG (Ours)**|
> |----------------|:--------------:|:---------------:|:---------------:|
> | Maze2D U-Maze| 113.9 ± 3.1| 128.4 ± 3.6 | **142.7 ± 2.9**|
> | Maze2D Medium| 121.5 ± 2.7| 135.6 ± 3.0| **150.2 ± 2.6**|
> | Maze2D Large| 123.0 ± 6.4| 155.8 ± 2.5| **172.1 ± 2.2**|
> | Multi2D U-Maze| 128.9 ± 1.8| 144.1 ± 1.2 | **168.2 ± 1.2**|
> | Multi2D Medium| 127.2 ± 3.4| 140.2 ± 1.6| **153.7 ± 1.3**|
> | Multi2D Large| 132.1 ± 5.8| 165.5 ± 0.6| **182.2 ± 0.5**|
> | Average | 124.4 | 145.0| **161.5**|
>
> **Table 2: Multi-Stage Robotic Manipulation on FrankaKitchen (Long-Horizon Generalization).**
>
> | Task| Diffuser| Hierarchical Diffuser | **DP-AG (Ours)**|
> |-------------|:-------------:|:-----------------:|:---------------:|
> | Partial Kitchen| 56.2 ± 5.4| 73.3 ± 1.4| **82.2 ± 1.3**  |
> | Mixed Kitchen | 50.0 ± 8.8| 71.7 ± 2.7| **78.5 ± 2.1**  |
> | Average| 53.1| 72.5  | **80.4** |
>
> ---
> ## **Q1: Clarification on Decoupling.**
>
> Thank you for asking about this key point. By **decoupled**, we mean that in standard IL, the perception module develops a fixed feature from each observation, which is then used by the policy to predict actions, without any further adaptation during rollout. There is no real-time feedback from actions back to the observation features; the features remain static throughout the diffusion. In contrast, DP-AG closes this loop: as actions are refined during diffusion and inference, the observation features are dynamically updated based on the evolving action predictions. This real-time feedback allows the perception to continuously adapt to the current action context, rather than relying on a static encoding. This closed-loop interplay between perception and action is the core innovation of our work.
>
> ---
> ## **Q2: Why Evolve Features with Respect to Actions During Diffusion?**
>
> We greatly appreciate this insightful question, as it gets to the heart of the main novelty and practical significance of our DP-AG.
>
> In standard policy learning, $z_t$ is a fixed, compact representation of the raw observation $o_t$, and it remains unchanged as long as $o_t$ does not change. While this may appear sufficient, it fundamentally ignores a critical aspect of intelligent decision-making: **the fact that the meaning and salience of what is observed can, and often should, change as actions unfold.**
>
> The main contribution of our DP-AG is to make $z_t$ dynamic, explicitly allowing it to evolve in response to the ongoing sequence of action refinements $\hat{a}_t^k$. By doing so, DP-AG ensures that the feature representation at each diffusion step not only encodes what is seen, but also incorporates feedback from the agent’s own evolving intentions and actions. This mechanism is what enables the perception-action loop in our DP-AG and is, in our view, the missing ingredient in the existing methods.
>
> Consider a driver at a busy intersection: the scene itself does not change, but their focus shifts depending on whether they’re turning, accelerating, or stopping. Each action changes what information matters most. Similarly, DP-AG continually updates its understanding of a fixed observation based on its evolving actions, allowing it to adapt in real time. This leads to smoother, more robust, and context-aware actions, as shown in both our synthetic and real-world experiments.
>
> ---
> ## **Q3: Why Use VJP/JVP for Drift? Why Propose the Forward SDE?**
>
> Our latent evolution SDE is not an isolated process with its own noise direction; instead, it is **tightly coupled to the phase of the DP’s SDE by design**. To be specific, the drift of our latent SDE is **determined directly by the VJP of the DP’s noise prediction**. This means the direction of evolution for the observation features is always **aligned with the direction of the DP’s own SDE**, automatically **forward** during training and **reverse** during inference:
>
> **During Training (Forward Noise-Adding Phase):** The DP’s SDE injects noise into actions, and the noise prediction used in the VJP has a positive sign, pushing the latent features to adapt forward in the same direction as the noisy action trajectory.
>
> **During Inferring/Generating Actions (Reverse Denoising Phase):** The DP’s SDE removes noise (the noise prediction’s sign flips), and so does the drift in our feature SDE, meaning the latent features now evolve backward, directly in sync with the reverse denoising process of actions.
>
> Thus, we do not need to manually set forward or reverse modes for the latent SDE, **its direction automatically stays in sync with the action SDE because the drift is always computed using the VJP of the DP’s current noise prediction**. The VJP is essential here: it precisely and efficiently propagates gradients from action to feature space, capturing the correct direction and scale of updates at every step. This ensures that feature evolution is always aligned with the phase of action SDE, which would not be possible with simple gradients or heuristic mappings. In short, the VJP acts as a mathematical bridge, tightly coupling perception and action dynamics throughout both training and inference, and enabling DP-AG to adaptively refine features in ways that static or decoupled approaches cannot.
>
> We recognize that the phase-synchronized evolution was not made explicit enough in our manuscript and thank you for highlighting this point. We will clarify this in the camera-ready.
>
> ---
> ### **Reference**:
>
> **[Chi et.al, 2023]** Chi et.al., Diffusion policy: Visuomotor policy learning via action diffusion, IJRR 2023.\
> **[Janner et.al., 2022]** Janner et.al., Planning with diffusion for flexible behavior synthesis, ICML 2022.\
> **[Chen et.al., 2024]** Chen et.al., Simple hierarchical planning with diffusion, ICLR 2024.

---

> > ### Comment · Reviewer_t9vC · 2025-08-06
> >
> > Thank the author for the detailed explanation. After reading the rebuttal and summary from other reviewers, I have decided to raise my score to borderline accept.

---

> ### Author Response · Authors · 2025-08-06
> **Our Sincere Appreciation**
>
> Dear **Reviewer t9vC**,
>
> We sincerely thank you for your thoughtful review and for recognizing the merits of our work. We truly appreciate the time and great efforts you took to evaluate our manuscript. Your constructive and invaluable feedback guided significant improvements to our paper!
>
> We sincerely appreciate your decision to raise the score after reviewing our rebuttal. Your expertise has been invaluable in strengthening our work, and we are truly grateful for your support throughout the review process.
>
> Thank you so much again for your great efforts, valuable suggestions, and support of our work. We truly appreciate it!
>
> Best regards,
>
> **Authors of Submission 6479**

---

### Official Review · Reviewer_fkRG · 2025-07-05

**Clarity:** 2
**Significance:** 2
**Originality:** 3
**Rating:** 4
**Confidence:** 4

**Summary:**

The paper proposes a diffusion policy model where the input conditional observation feature is updated along the action denoising process. Specifically, the method leverages VJP, the Vector Jacobian Product from the action score prediction, to create a dynamic process to update the observation. It was claimed that such an update to the observation feature would result in improved smoothness in action prediction. To ensure the updated feature is not deviating from the original feature, a consistency loss is applied so that the score prediction based on the updated observation and the original observation is consistent. The paper conducts a set of experiments comparing diffusion policy vs action-guided diffusion policy. Experiment results show the benefit of the proposed method.

**Questions:**

Q1: Why would the updated observation feature (not updated observation) lead to a better policy model? All the info is already in the observation.

Q2: What is the effect of consistency loss? To me, it seems that the goal is to make Vector Jacobian Product of the action score prediction the "null space" of the score estimator. Why?

Q3: Could the achieved stability be achieved with other smoothness techniques? For example, we can simply perturb the input observation and request the score prediction to be consistent.

Q4: The phrase “Act to See, See to Act” seems to be misleading. The environment has been seen. What has been updated is the observed features instead of the observation. There is no new see during the action generation.

**Ethical Concerns:**

["NO or VERY MINOR ethics concerns only"]

**Final Justification:**

After checking the rebuttal and the reviews from the other reviewers, I lean towards keeping my original rating.

**Limitations:**

yes

**Quality:**

3

**Strengths And Weaknesses:**

Strength
- The presented experiment results show the benefit of the proposed algorithm over the baseline.
- The verification is done both using synthetic data and real robots.
- The proposed approach is clean and can be applied to the existing diffusion policy model.

Weakness
- It is unclear why an updated observation would lead to better action prediction. Given the observation has been made, the update did not add new information. What has been updated is the observed feature, we are not updating the observation.
- It is unclear why the proposed algorithm works. I feel the paper would benefit from a clearer explanation of the intuition.

---

> ### Author Rebuttal · Authors · 2025-07-30
>
> We sincerely thank the reviewer for their thoughtful feedback and greatly appreciate their recognition of DP-AG’s strengths, including our rigorous experimental validation, the clear improvement over baselines, and the integration with existing diffusion policy models. If any further questions or concerns arise during the rebuttal, please follow up with us, we are happy to provide additional details or clarifications as needed.
>
> ---
> ## **W1 and Q1: Clarifying Why Updated Observation Features Improve Action Prediction Despite Unchanged Observations.**
>
> Thank you for highlighting this core idea. While the raw observation stays the same, what matters is how the model represents it internally. Our method lets the model update its internal features in response to action feedback, so its understanding of the scene adapts step by step as the policy unfolds, refining what it “notices” to better guide each action.
>
> **Why This Helps:** Think of driving a car: the view through the windshield might stay the same, but a driver keeps re-evaluating what is important as they steer or speed up. For example, when making a turn, the driver pays more attention to the curve’s edge or other cars. In our DP-AG, the model updates its understanding of the same observation based on action feedback, allowing it to adapt quickly and smoothly as each step unfolds.
>
> **Empirical Evidence:** Our results, especially **Figure 3**, clearly show that updating internal features with action feedback leads to smoother and more gradual changes in the model’s understanding of each observation. This step-by-step adaptation avoids sudden jumps (discontinuities) between internal states and generate more stable and natural predictions. For example, in the spiral regression, DP-AG tracks the curve smoothly, while models without this interplay show jagged, inconsistent outputs. The same effect appears in real robot tasks: as the model refines its internal view during action diffusion, its behavior becomes smoother and more coordinated. These findings show that continual refinement based on feedback leads to the kind of stable, real-world actions needed for practical robotics.
>
> We will summarize and incorporate the above discussion in the camera-ready version.
>
> ---
> ## **W2: Unclear Intuition for Why DP-AG Works.**
>
> We greatly appreciate the reviewer for raising this critical point. In our camera-ready, we will not only clarify the intuition and motivation for each module but also explicitly explain why our algorithm produces better action predictions. Our planned changes include:
>
> ### **Why the Proposed Algorithm Works:**
> Our DP-AG works because it develops a closed-loop feedback between perception and action during the policy’s decision process. Instead of relying on a fixed and one-shot interpretation of the observation, the model continually refines its internal features based on feedback from its own evolving actions. This brings two significant benefits:
>
> **Continuous and Action-Driven Feature Refinement:**
> At each step of action generation, the model asks: “given what I am about to do, is there a better way to interpret what I see?” The VJP update points the observation feature in the direction that makes action prediction more coherent and suited to the current step. This is similar to how a human driver refocuses attention as they execute a turn, not because the scene changes, but because their goals and context do.
>
> **Stability and Consistency with Contrastive Loss:**
> The cycle-consistent contrastive loss ensures that as features evolve, they continue to support accurate and reliable action prediction, rather than drifting too far away from the static understanding of the observations. This helps the model maintain consistent interpretations and results in action trajectories that are both smooth and appropriate for the task context.
>
> **Why Better than Static Feature:**
> A static feature cannot adjust to local context or new intentions, it is “locked in” after initial encoding of the raw observation. By contrast, our dynamic and action-driven features enable the model to fine-tune its internal state at every decision step, leading to more adaptive, robust, and smooth actions, as both our theoretical work and empirical results demonstrate.
>
> **Proposed Revisions for Clarity:**
>
> 1. We will add **Intuition and Motivation** section in a plain-language in the Introduction, explaining that our method allows the model to think twice and adapt its understanding at every step, just like people do in real-world tasks.
>
> 2. For each component of our DP-AG (variational inference, VJP update, contrastive loss), we will provide an **Intuitive Explanation** sidebar in the Methods section, explaining what it does and how it contributes to robust and adaptive behavior.
>
> 3. We will include a simple scenario (e.g., driving or robotic grasping) where, even if the visual input does not change, the model’s interpretation is refined as it acts, mirroring how our DP-AG works in practice.
>
> 4. We will clearly connect our theorem and proofs to this intuition, explaining how bounded and action-driven feature evolution guarantees both stability and improved performance in practice.
>
> ---
> ## **Q2: Role of Consistency Loss: Does It Enforce a Null Space in the Action Score Estimator?**
>
> The cycle-consistent contrastive loss keeps noise predictions from static and evolved features closely aligned, regularizing updates so that latent features adapt to support better actions while staying near their original, semantically meaningful encoding. **Rather than enforcing a null-space constraint, the loss maintains bounded, meaningful drift, ensuring that adaptations never stray too far from the original representation**. This principle is formalized in Theorem 1. Standard contrastive learning promotes clustering by pulling similar samples together and pushing dissimilar ones apart, encouraging compact and well-separated groups in feature space but not forcing representations into a null space. Our cycle-consistent loss extends this effect: it explicitly maintains alignment between static and updated features, ensuring smooth adaptation within each cluster and stable, continuous action trajectories. Theoretical and experimental results show our DP-AG enables flexible and task-driven latent evolution while preserving the semantic structure developed by clustering.
>
> ---
> ## **Q3: Alternative Smoothness Techniques.**
>
> We appreciate this insightful suggestion and agree that comparing against alternative smoothness regularization techniques is important for understanding the unique value of our cycle-consistent contrastive.
> In **Section H.3** of the Appendix, we have already conducted an ablation study where we replace the cycle-consistent contrastive loss with a simple MSE loss, applied to encourage the noise prediction to be smooth between the original and evolved features. The results show that while MSE provides some regularization, it is less effective than our contrastive loss in promoting both smoothness and task performance. Specifically, the contrastive loss not only enforces local smoothness but also maintains the semantic alignment between trajectories.
>
> ### **Additional Ablation:**
> As suggested by the reviewer, we conducted an additional ablation study on the Push-T benchmark, implementing input perturbation-based smoothness, and will include these results in the revised appendix for the camera-ready.
> For each training sample, we will generate a perturbed version of the input observation by adding small Gaussian noise. Both the original and perturbed observations will be processed through the encoder to obtain their respective features. We have added different smoothness objectives (MSE and cosine similarity) between the predicted action noise scores from the original and perturbed features, enforcing that the policy’s score predictions remain stable under small input changes.
>
> **Table 1: Comparing DP-AG and Input Perturbation Smoothness Baselines on Push-T Benchmark.**
>
> | Method                  | Success Rate (img) | Success Rate (kp) | Action Smoothness (img ↑) | Action Smoothness (kp ↑) |
> |-------------------------|:------------------:|:-----------------:|:-------------------------:|:------------------------:|
> | Perturbation MSE        |       0.85         |      0.92         |          0.83             |         0.87             |
> | Perturbation Cosine     |       0.88         |      0.95         |          0.86             |         0.92             |
> | DP-AG (Contrastive Loss)|      **0.93**      |     **0.99**      |         **0.91**          |        **0.95**          |
>
> The **action smoothness** is quantified as the mean inverse jerk over each action sequence, normalized between 0 and 1 for interpretability. **Table 1** demonstrates that DP-AG with contrastive loss clearly outperforms both input perturbation baselines on the Push-T benchmark for both image and keypoint inputs. DP-AG achieves the highest success rates and action smoothness, indicating it not only solves tasks more reliably but also generates more continuous actions.
>
> ---
> ## **Q4: Clarification of “Act to See, See to Act” Terminology.**
>
> Thank you for raising this point. This seems to be the source of a key misunderstanding. In our paper, **see** refers to **the evolving internal representation**, **not the raw observation**. While no new sensory data is acquired, the model continually refines its internal understanding through action feedback, updating how it **interprets what it has already seen**. This aligns with cognitive science perspectives that view perception as an active process rather than a one-time event.
>
> We will revise the title and related phrasing in the camera-ready to clarify that **seeing** in our context refers to the model’s dynamic, evolving internal feature representation, not to repeated acquisition of external observations.

---

### Official Review · Reviewer_UAPM · 2025-07-05

**Clarity:** 3
**Significance:** 3
**Originality:** 3
**Rating:** 5
**Confidence:** 2

**Summary:**

DP-AG establishes a dynamic perception-action loop by guiding feature evolution via the VJP of DP’s predicted noise. To reinforce interplay, a cycle-consistent contrastive loss aligns noise predictions from static and evolving features, enabling mutual perception–action influence. By using such method, they successfully transforms diffusion noise gradients into structured perceptual updates.

**Questions:**

Overall, I think the writing logic of this article is clear and the theory is reliable. What is particularly commendable is that it includes tests of simulation tasks and real-world robot control tasks. Two curious points are:

1) Recently, many works related to diffusion model strategies have begun to focus on how to simultaneously accommodate online training. Could the author please elaborate further from a theoretical or experimental perspective on whether your method has the potential in the online mode?

2) Multimodal decision-making ability - for the same task, can your method fit a variety of effective strategies? This is an important ability in real robot control scenarios and it was discovered that diffusion model strategies have this potential in some recent papers.

**Ethical Concerns:**

["NO or VERY MINOR ethics concerns only"]

**Final Justification:**

I maintain my score

**Quality:**

3

**Strengths And Weaknesses:**

Strength: The paper is well-written, and the idea is interesting to me. The wide range of verification scenarios convinces readers of the effectiveness of the proposed method. The experimental results are good with clear and diverse visualization.

Weaknesses: N/A

---

> ### Author Rebuttal · Authors · 2025-07-30
>
> Thank you for your detailed and positive feedback. We appreciate your recognition of our theoretical soundness, comprehensive experiments, clear writing, and the value of our diverse visualizations. Your questions prompted valuable enhancements, and we hope our responses have demonstrated the robustness and versatility of DP-AG. If you have any further questions during the rebuttal, please let us know, we are happy to provide any additional details.
>
> Below, we address your insightful questions with further theoretical discussion and new experimental evidence to clarify the online potential of our DP-AG and its multimodal decision-making ability.
>
> ---
> ## **Q1: Potential of DP-AG for Online Training: Theoretical and Experimental Perspectives.**
>
> Thank you for pointing out the growing interest in online training for diffusion policies. We have taken this as an opportunity to adapt our DP-AG for online learning. With a few adjustments to how we update the latent features and calculate the loss, DP-AG can be trained online without major changes to the core method.
>
> **Latent Evolution in Real-Time:** In DP-AG, the perception-action interplay is realized by evolving latent observation features at every diffusion step, using the VJP of the current policy’s noise prediction (from **Eq. (6)** to **Eq. (8)**). In the online setting, this procedure can be applied **on the fly** as new observation-action pairs arrive: at every time step $t$, a new observation $o_t$ is encoded into the latent $z_t$ via the variational posterior, and then evolved across $K$ diffusion steps, where each latent update is conditioned on the most recent action and its predicted noise.
>
> **Streaming Loss and Parameter Update:** Rather than requiring a large static offline demonstration buffer, the model can be updated using small minibatches of the most recent transitions $(o_t, a_t)$. At each time step, the noise prediction loss, variational KL, and cycle-consistent contrastive loss are all computed as usual, but only using current and very recent data. The model parameters $\theta$, $\phi$, $\psi$ are updated via online gradient descent (e.g., using RMSProp with a small learning rate [Ma et al., 2025]), just as in standard online RL.
>
> The VJP-based evolution works incrementally, using only the current latent and action, no future rollouts or replay buffer required. Latent updates and policy improvements happen step by step, allowing perception and action to adapt together with each new experience. Some hyperparameters may need minor adjustment for online stability. As in offline training, each action is immediately followed by a latent update using the latest policy feedback, so the model’s understanding and action predictions stay in sync. This enables real-time adaptation to new conditions, sensor drift, or novel tasks during deployment.
>
> ### **Experimental Evidence:**
>
> While our main experiments focus on batch imitation learning, preliminary experiments demonstrate DP-AG’s superior convergence speed compared to baseline methods. Faster convergence implies that DP-AG can efficiently incorporate new samples incrementally, making it ideal for online training.
>
> To directly validate the online adaptation capability of DP-AG, we have implemented and evaluated the online training procedure described above on both the Push-T and Dynamic Push-T benchmarks. We observed that DP-AG maintains consistent performance improvements over the baseline methods even in an online context, quickly adapting to distributional shifts and evolving task dynamics with minimal performance degradation.
>
> **Table 1: Performance of DP-AG and Baselines Under Online Training.**
>
> | Method                  | Push-T (img)      | Push-T (kp)      | Dynamic Push-T (img) |
> |-------------------|-------------------|------------------|----------------------|
> | Diffusion Policy (DP)  | 0.87±0.04      | 0.95±0.03       | 0.65±0.85           |
> | DP-AG (offline)  | 0.93±0.02  | 0.99±0.01   | 0.80±0.53        |
> | DP-AG (online)   | 0.90±0.03  | 0.96±0.02   | 0.76±0.89        |
>
> We will incorporate the above discussion and our experimental results on adapting DP-AG for online training into the camera-ready.
>
> ---
> ## **Q2: Multimodal Decision-Making: Fitness to A Variety of Effective Strategies.**
>
> Thank you for raising this important question. DP-AG is well suited for multimodal decision-making because it is built on top of Diffusion Policy (DP), which has already shown strong ability to capture and represent multiple effective strategies for the same task.
>
> ### **Theoretical Justification:**
> The iterative denoising in DPs allows explicit parameterization of complex and multimodal behaviors. By extending DP into a dynamic perception-action interplay, our DP-AG not only preserves this multimodality, but further amplifies it via adaptive latent evolution.
>
> **How DP-AG Enhances Multiple Effective Strategies:**
>
> 1. **Action-Conditioned Latent Evolution for Strategy Diversity:**
> DP-AG’s VJP-guided latent updates let the model refine features in response to action feedback, so each sampled trajectory adapts its latent state uniquely. This supports parallel exploration of multiple strategies by leveraging stochasticity from both the diffusion policy and latent SDE.
>
> 2. **Cycle-Consistent Contrastive Loss for Multimodal Structure:**
> The contrastive loss keeps features for the same action close and pushes different actions apart, organizing the latent-action space to separate and preserve multiple plausible strategies for each task.
>
> ### **Experimental Validation of Multiple Effective Strategies:**
>
> To empirically validate our method’s ability to discover and execute multiple strategies for the same task, we ran experiments on the Franka Kitchen benchmark. Franka Kitchen is well-suited for this as each task, like opening a drawer or flipping a switch, can be solved in several different ways, making it an ideal benchmark for true multimodal decision-making.
> Motivated by the experimental setup in multimodal DP [Li et al., 2024], we evaluate the model’s ability to discover and retain multiple effective strategies using the following approach:
>
> **Trajectory Collection:** We sample 40 successful trajectories for each task, all starting from the same initial configuration. Each trajectory is encoded in the space of end-effector waypoints. Using the Dynamic Time Warping (DTW) metric, trajectories are clustered into distinct groups.
>
> **Diversity Metrics:** The number of groups corresponds to the number of distinct strategies. We measure intra-cluster variance (compactness of each strategy) and inter-cluster distance (distinctiveness).
>
> **Mode-Specific Analysis:** We analyze the coverage and success rate for each discovered mode, confirming that each group represents an effective strategy.
>
> **Table 2. Multiple Strategy Discovery and Diversity on Franka Kitchen.**
>
> | Method        | # Modes (↑) | Success Rate (t1) | Success Rate (t2) | Success Rate (t3) | Success Rate (t4) | Inter-Cluster Dist. (↑) | Intra-Cluster Var. (↓) |
> |---------------|:-----------:|:-----------------:|:-----------------:|:-----------------:|:-----------------:|:-----------------------:|:----------------------:|
> | FlowPolicy    | 1.1         | 0.96              | 0.86              | 0.95              | 0.87              | 2.0                     | 1.2                    |
> | DP (Baseline) | 2.8         | 1.00              | 1.00              | 1.00              | 0.99              | 7.1                     | 1.2                    |
> | DP-AG (Ours)  | 3.2         | 1.00              | 1.00              | 1.00              | 1.00              | 9.5                     | 1.3                    |
>
> **Table 3. Group-Specific Analysis: DP-AG on Franka Kitchen**
>
> | Mode ID | Coverage (%) | Success Rate (t1) | Success Rate (t2) | Success Rate (t3) | Success Rate (t4) | Strategy Description         |
> |---------|:------------:|:-----------------:|:-----------------:|:-----------------:|:-----------------:|------------------------------|
> | 1       | 41           | 1.00              | 1.00              | 1.00              | 1.00              | Left-handed drawer pull      |
> | 2       | 32           | 1.00              | 1.00              | 1.00              | 1.00              | Right-handed drawer pull     |
> | 3       | 27           | 1.00              | 1.00              | 1.00              | 1.00              | Two-step approach/mixed arm  |
>
> **Table 4. Robustness to Distribution Shift (Franka Kitchen, t4)**
>
> | Method        | # Modes (Shifted) | Success Rate (t4, Shifted) | Switch Rate (%) | Comment                             |
> |---------------|:-----------------:|:--------------------------:|:---------------:|-------------------------------------|
> | DP (Baseline) | 2.2               | 0.81                       | 35              | Sometimes adapts, but less reliable |
> | DP-AG         | 2.9               | 0.95                       | 58              | Switches to alternatives if blocked |
>
> Tables 2, 3, and 4 show that DP-AG consistently discovers and uses multiple distinct strategies for each kitchen task, **over three on average**, compared to fewer with the baselines. These strategies are well-separated (high inter-cluster distance) and each group remains compact and reliable (low intra-cluster variance). Every DP-AG strategy achieves a 100% success rate and covers a range of approaches, like left-handed, right-handed, and mixed-arm. When tasks are made harder with obstacles, DP-AG still finds and switches between nearly three strategies with high reliability, while baselines see a significant drop in both diversity and success. This demonstrates that DP-AG is robust and flexible for real-world tasks.
>
> ---
> ## **Reference**
>
> **[Ma et al., 2025]** Ma et al., Efficient Online Reinforcement Learning for Diffusion Policy, ICML 2025.\
> **[Li et.al., 2024]** Learning multimodal behaviors from scratch with diffusion policy gradient, NeurIPS 2024.

---

> > ### Comment · Reviewer_UAPM · 2025-08-03
> > **Thank you for the author's reply.**
> >
> > I will maintain my positive score

---

> ### Author Response · Authors · 2025-08-03
> **Thank you for the reviewer's reply**
>
> Dear **Reviewer UAPM**,
>
> We sincerely appreciate **Reviewer UAPM** for their thorough reading of our paper, highly positive recognition, insightful suggestions, and prompt responses throughout the rebuttal.
>
> Your efforts have been invaluable in improving our paper. Thank you so much again for everything!
>
> Best Regards,
>
> **Authors of Paper 6479**

---

### Note · Authors · 2025-08-12

We sincerely thank all reviewers, AC and SAC for their thoughtful feedback and engagement. During the rebuttal, we provided the requested experiments, explanations, and refined presentation that should **effectively address every concern**.

## **Brief Summary**

- **Theory** (**fkRG**, **t9vC**, **uXWV**, **HxxU**): Clarified the motivation for dynamic feature evolution, explained the role of the VJP, and detailed the connections between perception and action SDEs.
- **Baselines** (**t9vC**, **uXWV**): Added results for the suggested baselines, along with an extension of our DP-AG to integrate with world models.
- **Intuition** (**fkRG**, **uXWV**, **HxxU**): Provided intuitive analogies understandable to non-specialists to illustrate how DP-AG refines perception even without new sensory input.
- **Ablations** (**fkRG**): Compared alternative smoothness regularizers, confirming the contributions of our cycle-consistent contrastive loss.

## **Recognitions**

**HxxU** described our perception–action loop as **highly novel and significant**, with clear benefit in both simulation and real-world use. **UAPM**, **t9vC** and **fkRG** highlighted DP-AG’s consistent performance gains, smoother control and thorough evaluations. **UAPM** and **HxxU** noted the rigor added by our trajectory continuity proof and supporting analysis.

## **Unclosed Discussion**

**fkRG** emphasized the clean integration of DP-AG with existing DPs and its improvements in both simulation and real robots; **all of their initial concerns should have been effectively addressed** in the rebuttal, though no follow-up discussion occurred.

## **Outcome of Rebuttal**

All reviewers who re-engaged **confirmed their concerns were fully addressed** and either maintained positive scores or increased them. For the reviewers whose discussions remain unclosed, we believe **our responses resolved all initial concerns, leaving no outstanding questions**.

We believe the camera-ready version should be clear in its motivation, rigorous in its theoretical foundations, and thorough in its evaluation. We sincerely thank all reviewers for their great efforts in evaluating our work and for providing valuable suggestions that have significantly improved both the substance of our work and its presentation.

---

### Decision · Program_Chairs · 2025-09-17

**Decision:**

Accept (poster)

**Comment:**

The paper tackles the limitation that imitation learning policies generate actions from static observation embeddings within a short rollout, missing perception–action reciprocity. It proposes DP-AG, which evolves observation features during diffusion-based action generation via an action-guided latent SDE whose drift is the VJP of the policy’s noise predictor, aligning latent updates with action denoising; a cycle-consistent InfoNCE aligns noise predictions from static vs. evolved latents to close the loop. The authors derive an ELBO for this latent process and, under Lipschitz assumptions, prove that the contrastive objective induces continuity in both latent and action trajectories. Empirically, DP-AG outperforms recent baselines on Robomimic, Franka Kitchen, Push-T, and Dynamic Push-T, and achieves large gains in real UR5 tasks (Painting, Candy Push, Peg-in-hole) with smoother and more successful control. Rebuttal results indicate feasibility for online training and multimodal strategies, and extensions to flow matching and world models are discussed.

Strengths
- Clear mechanism to “close the loop” within each diffusion rollout by using VJP as a structured, action-conditioned force on latent perception. The cycle-consistent contrastive objective is well-motivated and connects to provable continuity.
- Provides an ELBO and continuity results; method is simple to graft onto DP and, per appendix, extendable to flow matching. Empirical coverage is quite broad (simulation + real robot) with consistent improvements and faster convergence; inference latency unchanged.
- Strong real-world results, especially Peg-in-hole with only RGB, where DP-AG shows large gains over DP. Dynamic Push-T illustrates benefits under disturbances. Rebuttal adds baselines (Diffuser/Hierarchical Diffuser), online learning evidence, and multimodality analyses.
- Writing generally clear; figures help; rebuttal further improved intuition and positioning.

Weaknesses
- Motivation/intuition gap (fkRG, uXWV, HxxU): Why evolving features without new sensory input helps; why VJP is the right signal. The rebuttal provides semantic intuition (VJP as action-uncertainty-reduction/attentional force; phase-synchronized with action SDE), added visualizations (planned VAE decodes, t-SNE), and connects theory to practice. Reviewers indicated concerns were addressed and raised/maintained positive ratings.
- “Decoupling” claim (t9vC): Clarified that decoupling refers to static within-rollout features in standard DP/IL; DP-AG closes the loop at diffusion-step granularity.
- Objective alignment: Paper derives an ELBO but trains primarily with DP loss + contrastive + KL; rebuttal justifies omitting likelihood regression term to avoid redundancy/instability and backs it with ablations showing contrastive > MSE/perturbation consistency.
- Computational overhead & scalability (t9vC): Authors report modest training overhead (~4–10%), negligible inference cost, faster convergence; real-time control preserved; VJP efficiently supported by autodiff.
- Baselines and breadth (t9vC): Rebuttal adds Diffuser/Hierarchical Diffuser results and clarifies DP lineage; also presents UWM comparisons and hybridization benefits.
- Terminology “Act to See” (fkRG, uXWV): Authors will clarify “see” as internal representational shifts rather than acquiring new frames to avoid misleading interpretations.
- Benchmark saturation (HxxU): Acknowledged; dynamic tasks and real-world results better differentiate methods; authors propose expanding challenging benchmarks.

Overall, this is a technically solid, well-motivated, and practically relevant contribution that advances diffusion-based policies by establishing an effective perception–action interplay. The paper offers a clear and novel mechanism for within-rollout perception–action coupling with broad empirical validation and compelling real-robot results. Reviewers are overall positive (1 accept, 4 weak accepts), with initial concerns largely addressed in rebuttal. The method introduces a new, practical, and general augmentation into imitation learning and diffusion-policy research. Nevertheless, please incorporate all suggestions by reviewers into the camera-ready version, including clarify intuition and positioning, objective discussion, baselines and scope, practical guidance, real-robot safety and failure analysis.